# To CoT or not to CoT? Chain-of-thought helps mainly on math and symbolic reasoning

**Zayne Sprague♠, Fangcong Yin♠, Juan Diego Rodriguez♠, Dongwei Jiang◇,
Manya Wadhwa♠, Prasann Singhal♠, Xinyu Zhao♠,
Xi Ye♡, Kyle Mahowald♠, Greg Durrett♠**

♠The University of Texas at Austin, ◇Johns Hopkins University, ♡Princeton University
zaynesprague@utexas.edu

## Abstract

Chain-of-thought (CoT) via prompting is the de facto method for eliciting reasoning capabilities from large language models (LLMs). But for what kinds of tasks is this extra "thinking" really helpful? To analyze this, we conducted a quantitative meta-analysis covering over 100 papers using CoT and ran our own evaluations of 20 datasets across 14 models. Our results show that CoT gives strong performance benefits primarily on tasks involving math or logic, with much smaller gains on other types of tasks. On MMLU, directly generating the answer without CoT leads to almost identical accuracy as CoT *unless* the question or model's response contains an equals sign, indicating symbolic operations and reasoning. Following this finding, we analyze the behavior of CoT on these problems by separating planning and execution and comparing against tool-augmented LLMs. Much of CoT's gain comes from improving symbolic execution, but it underperforms relative to using a symbolic solver. Our results indicate that CoT can be applied selectively, maintaining performance while saving inference costs. Furthermore, they suggest a need to move beyond prompt-based CoT to new paradigms that better leverage intermediate computation across the whole range of LLM applications [1] .

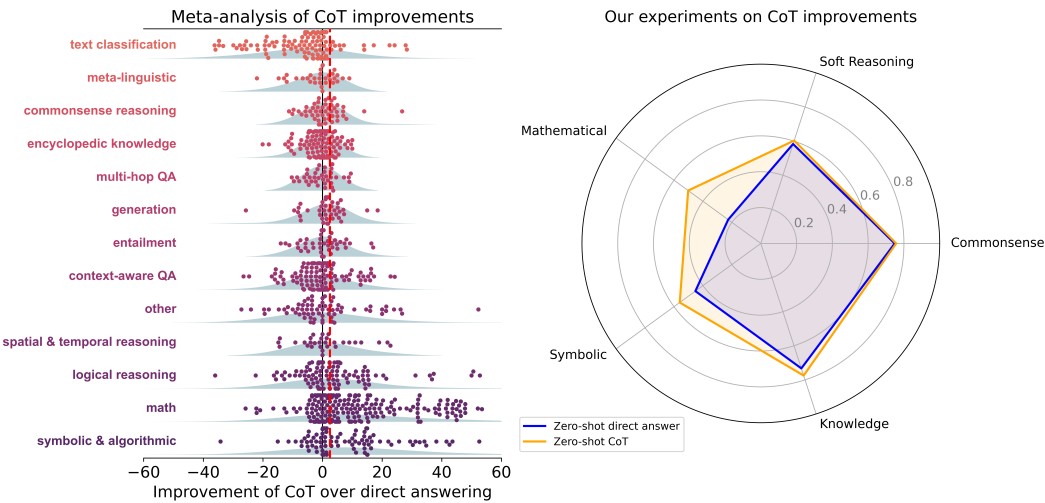

Figure 1: Left: meta-analysis of CoT literature; each point is a reported delta of CoT over direct answering for some (LLM, task) pair. Right: average performance of using zero-shot CoT v.s. direct answer prompts across five general reasoning categories, covering 20 datasets with 14 LLMs evaluated on each. In both sets of results, math and other kinds of symbolic reasoning are the domains that consistently see substantial improvements from CoT (red dotted line indicates the mean improvement from CoT across experiments).

---

[1] Our code can be found at https://github.com/Zayne-sprague/To-CoT-or-not-to-CoT.

## 1 INTRODUCTION

Chain-of-thought (CoT) (Nye et al., 2022; Wei et al., 2022) has become a widely used prompting technique for eliciting reasoning from language models. CoT can provide human-readable explanations of how problems are solved (Joshi et al., 2023; Lanham et al., 2023), but most frequently it is invoked to improve an LLM's ability to answer complex questions via intermediate computation (Madaan & Yazdanbakhsh, 2022; Wang et al., 2023a; Dziri et al., 2023). Current post-training schemes for LLMs heavily infuse CoT capabilities into models: systems like ChatGPT or Llama 3.1 default to CoT when given reasoning problems (OpenAI, 2023; Dubey et al., 2024).

CoT has seen widespread usage, but it is most heavily explored in the domain of mathematical reasoning (Zhou et al., 2023a; Fu et al., 2023; Chae et al., 2024; Xu et al., 2024b; Qi et al., 2024). In fact, many "reasoning" methods for LLMs are evaluated *only* in the math domain; for instance, Lightman et al. (2024) frame their paper as "complex multi-step reasoning" and Mixtral-Large2's release [2] cited effort "enhancing the model's reasoning capabilities", but performance is only reported on GSM8K and MATH. CoT is reported to be effective across a wide range of studies, but many of these studies focus on a narrow slice of the task space. In areas beyond math, results show that CoT is not as useful (Kambhampati et al., 2024a) or can even hurt performance (Wang et al., 2024).

In this work, we aim to evaluate where prompt-based CoT helps and why. We begin with a systematic meta-analysis of recent literature that reports performance of CoT versus direct answering (DA). We then augment this picture by conducting experiments on 20 datasets and 14 contemporary LLMs across zero-shot and few-shot prompt settings. **Finding 1: CoT only helps substantially on problems requiring mathematical, logical, or algorithmic reasoning.** Figure 1 shows this holds both across the literature and our own experiments. We find only a few cases of large gain in other kinds of tasks, and many of these outliers feature some component of symbolic reasoning. For instance, on MMLU (Hendrycks et al., 2021a) and MMLU Pro (Wang et al., 2024), we analyze the improvements from CoT and find that CoT *only* gives benefit on math slices of the dataset. **As much as 95% of the total performance gain from CoT on MMLU is attributed to questions containing "=" in the question or generated output.** For non-math questions, we find no features to indicate when CoT will help.

How can we better understand *why* CoT improves on these questions and only these questions? The math and formal logical reasoning datasets we consider can be broken down into two stages of processing: a planning step (e.g., parsing a problem into equations) and an execution step (building intermediate outputs and working towards a solution) (Ye et al., 2023; Wang et al., 2023b; Sun et al., 2024). **Finding 2: CoT primarily helps with the execution step that performs computation and symbolic manipulation, but falls short of what LLMs with tool augmentation can do.** We find that LMs prompted with CoT can generate executable formal solution plans and execute those plans better than direct answering. But using LMs to generate a solution plan and then using an external symbolic solver to solve the plan outperforms using CoT for both steps for these tasks.

These results paint a picture that CoT's utility is often circumscribed by tool augmentation: on problems where CoT helps, we already have more powerful tools than CoT that we can employ, and on "soft reasoning" problems like commonsense where no tools exist, we see limited benefit from CoT. This characterization has two major implications. First, CoT is unnecessary for many problems where it is widely employed: there exist more efficient prompting strategies that yield similar performance for much lower inference cost. Second, we see a critical need to move beyond prompt-based CoT to more sophisticated approaches based on search, interacting agents, or models more heavily fine-tuned for CoT. Future work can explore how intermediate computation can be better used to solve challenging problems outside of the math and symbolic reasoning domains.

## 2 BACKGROUND: CHAIN-OF-THOUGHT

The tasks we consider in this work consist of a question $\mathbf{q} \in \Sigma^*$ for a vocabulary $\Sigma$ and an answer $a \in \mathcal{L}(\mathbf{q})$ for a label set $\mathcal{L}(\mathbf{q})$. $\mathcal{L}(\mathbf{q})$ can consist of a data type like boolean or integer, classification labels, or problem-dependent labels like names of entities from $\mathbf{q}$. One exception that we still

---

[2]https://mistral.ai/news/mistral-large-2407/

explore is BiGGen Bench (Kim et al., 2024), which instead relies on an LLM-as-a-judge (Dubois et al., 2023; Zheng et al., 2024b) to provide a label for generated long-form responses.

**Prompting and chain-of-thought for reasoning**    A large language model places distributions over strings $p(\mathbf{y}) = \prod_{i=1}^{n} p_{\text{LM}}(y_i)$ where $\mathbf{y} \in \Sigma^*$. In practice, we can interpret these as conditional distributions $p(\mathbf{y} \mid \mathbf{x})$ where $\mathbf{x}$ is a user's prompt. Typical invocation of an LLM involves forming a prompt $\mathcal{I}(\mathbf{q})$ that wraps the question with additional instruction, then drawing a sample response $\tilde{\mathbf{y}} \sim p(\mathbf{y} \mid \mathcal{I}(\mathbf{q}))$, and finally returning $a = \text{extract}(\tilde{\mathbf{y}})$ using some kind of answer extractor.

For the tasks we consider in this work, the output $\tilde{\mathbf{y}}$ can take one of two forms. A **direct answer** only contains a string realization of $a$; e.g., $\mathbf{y} = (\_185, 4)$ which is detokenized as the answer $a = 1854$. A **chain of thought** is a longer sequence $\mathbf{y}$ including other tokens beyond the answer, e.g., $\mathbf{y} = (\_185, 6, \_\text{minus}, \_2, \_\text{equals}, \_185, 4)$. In both cases, the extract function must parse and detokenize the output; in CoT, there is some extra work to spot where the answer is placed.

Our prompts can explicitly encourage use of direct answer or chain of thought as strategies, which we denote as $\mathcal{I}_{\text{da}}$ and $\mathcal{I}_{\text{cot}}$. For eliciting CoT, this includes strategies like telling a model to "*think step by step*" (Kojima et al., 2022). For directly answering a question, a prompt may say "*immediately generate the answer*". We track the average location of the answer in the generated output for both CoT and direct prompts in Appendix F.3 to ensure that direct answer prompts give the answer early in the output. We also ensure that extract can parse answers from the generated output for each model, prompt, and dataset combination used in our experiments, tailoring the extract function as needed to ensure low unparseable rates for each model and task.[3] All prompts and outputs per dataset per model have been uploaded to Huggingface and we include examples of some of our prompts in the Appendix J. We also experiment with few-shot CoT prompts, which we find perform similarly to zero-shot prompts; details about these are given in Appendix E.

**Symbolic reasoning**    Of key importance to this work is whether problems feature symbolic reasoning or not. We consider a problem to be **symbolic** if it can be grounded in a *natural, well agreed-upon* formal system. "$12 \times 4$" is an example of a symbolic problem, which can be grounded in mathematics. Other systems include first-order logic (Saparov & He, 2023; Hua et al., 2024) or planning languages (Liu et al., 2023a; Valmeekam et al., 2023). Formally, for symbolic problems, we define a function $f$ that acts as a map that produces some symbolic expression $\mathcal{S} = f(\mathbf{q})$ from the question. $\mathcal{S}$ can be used as input for a solver to derive an answer, $\hat{a} = \text{solve}(\mathcal{S})$.

Conversely, a problem like *where on a river can you hold a cup upright to catch water on a sunny day?* from CommonsenseQA (Talmor et al., 2019) is **non-symbolic** by our definition. While this problem could be formalized with some kind of predicate logic (Zhou et al., 2022; Quan et al., 2024; Zhou et al., 2024) or grounded in some kind of physical simulation (Hao et al., 2023; Wong et al., 2023), there is not a natural nor well agreed-upon framework for solving it.

We view non-symbolic to symbolic reasoning as a spectrum. MuSR (Sprague et al., 2024) is a "semisymbolic" dataset in that it does contain an underlying formal system (e.g., for its murder mysteries portion, the notion that $\text{motive}(X) \wedge \text{means}(X) \wedge \text{opportunity}(X) \implies \text{murderer}(X)$), but also involves substantial commonsense reasoning that does not map onto a formal system. In these cases, we can still form $\mathcal{S} = f(\mathbf{q})$, but $f$ must rely heavily on a language model and instantiate new information for $\mathcal{S}$ that is not directly represented in $\mathbf{q}$.

**Central claim**    Figure 1 shows that there are a large number of positive results on CoT reported in the literature. Informally, we believe many readers of the literature to hold the following view: $\mathcal{I}_{\text{cot}}$ *will outperform $\mathcal{I}_{\text{da}}$ on nearly all reasoning problems, whether those problems involve symbolic or non-symbolic reasoning.* Our evidence does *not* support this conjecture. We will show that this performance boost is strongest for symbolic and semi-symbolic tasks, while giving little to no improvement (or even hurting performance) on non-symbolic tasks.

---

[3]We exclude a number of other "CoT-like" approaches in our analysis such as decomposed prompting (Khot et al., 2023; Zheng et al., 2024a) and multi-agent debate (Du et al., 2023; Chen et al., 2024). We focus on single prompt approaches. We deal with tool-augmented approaches in Section 5.

Table 1: A few categories for experimental comparisons. Full list in Appendix B.

| Category | Description |
|---|---|
| Symbolic and algorithmic | Tasks involving symbol manipulation which can be solved by executing a program. This includes entity tracking datasets (e.g., SCONE, Coin Flip) and algorithmic tasks (e.g., BBH word sorting or finding shortest paths in a graph). |
| Math | Tasks requiring mathematical reasoning, from grade-school math to advanced mathematics, including physics questions. |
| Logical reasoning | Tasks designed to test for logical reasoning, whether deductive (Saparov & He, 2023, PrOntoQA), inductive (Bowen et al., 2024) or analogical (Ma et al., 2024) reasoning, including syllogisms and logical puzzles. |
| Encyclopedic knowledge | Tasks requiring expert-level in-depth knowledge beyond mere common-sense, usually in an open-book setting. |
| Mixed datasets | Datasets containing a variety of tasks, such as BIG-Bench Hard (BBH) or MMLU. |
| … | … |

## 3 RESULTS FROM THE LITERATURE

**Criteria and Process**    We investigate all papers from ICLR 2024, a representative ML venue, and two representative NLP venues, EACL 2024 and NAACL 2024 (including Findings and Workshop papers). This resulted in 4,642 papers total that filtered using automatic and manual methods to papers including experiments comparing chain-of-thought, $\mathcal{I}_{\text{cot}}$, against direct answering prompts, $\mathcal{I}_{\text{direct}}$. A total of 110 papers were found that matched our criteria with 1,218 experimental comparisons. We then grouped the comparisons by the types of tasks and datasets being evaluated. More details on our automatic and manual filtering, as well as our categorization, can be found in Appendix A and B.

**Results**    Figure 2 shows the distribution of CoT deltas (CoT prompt minus the direct answer prompt performance) across our categorization of different task types found in the literature. Compared to Figure 1, we take the mean results per paper per category, indicated by blue dots, showing the trend across papers in the literature. The categories are ranked in order of ascending median CoT delta. The three categories which benefited the most from CoT are symbolic reasoning, math, and logical reasoning, with average improvements of 14.2, 12.3, 6.9, respectively. Average performance on these top three tasks with CoT was 56.9, whereas performance without CoT was 45.5. For other categories, the average performance with CoT was 56.8, compared to 56.1 without CoT. We do not consider this small improvement a victory for CoT. CoT involves more computation than direct answering, and a truly fair comparison between the methods should match the compute of the two methods, e.g., ensembling across multiple prompts.

**Do any non-math datasets benefit from CoT?**    On the right side of Figure 2, we show the top 10 outliers from our observed trend, namely papers with high CoT deltas averaged across experiments in tasks *other than* math, symbolic, or logical reasoning. Although not categorized as math or logic, several of these are related to logical, mathematical or symbolic reasoning in some way. From this list, the dataset which benefits the most most from CoT is BIG-bench Hard (BBH) (Suzgun et al., 2023), a benchmark consisting largely of problems requiring algorithmic, arithmetic or logical reasoning. For instance, BIG-bench Navigate is a spatial reasoning task, but relies heavily on a mathematical primitive of counting steps taken to derive a final conclusion. Similarly, while BIG-bench Temporal is a temporal reasoning task (answering questions about when certain events could have occurred), it requires deductive reasoning to solve. In addition, Legal Argument Reasoning (SemEval-2024 Task 5) (Bongard et al., 2022) was categorized as *context-aware QA*, but also requires substantial reasoning ability. Finally, MMLU-Moral Scenarios (Hendrycks et al., 2021a) requires answering two independent questions at once, which essentially involves a symbolic combination of two simpler questions.

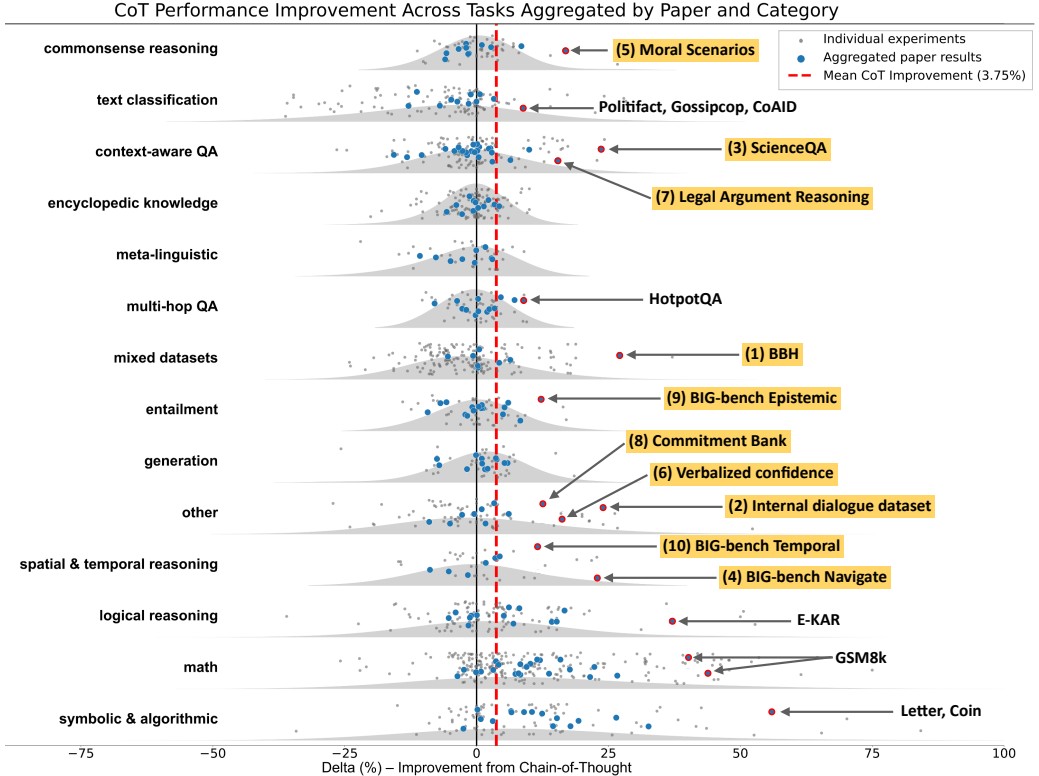

Figure 2: Results from our meta-analysis (grey dots) aggregated by paper and category (blue dots).

There are a few outliers that less clearly follow the trend. ScienceQA (Lu et al., 2022) consists of multiple choice questions across a range of natural and social science disciplines, though it is hard to interpret gains without knowing breaking down performance by subject or question type. The dialogue evaluation dataset from Jia et al. (2024) sees large improvements with CoT, but this is a proprietary dataset, and we note that other essay scoring results in our meta-analysis (Li et al., 2024; Stahl et al., 2024) did not show improvements with CoT. Other non-math, symbolic or logical datasets that benefit from CoT are Commitment Bank (de Marneffe et al., 2019) and the task of eliciting verbalized confidence (Xiong et al., 2024). Nevertheless, these are exceptions to the rule. The majority of the reported benefits from using CoT in the NLP and ML literature comes from math or math-related tasks.

## 4 RESULTS FROM EXPERIMENTS

### 4.1 EXPERIMENTAL SETUP

**Dataset, Models, Prompts** All datasets, models, and prompts we evaluate over can be found in detail in the tables 3, 4, and 5 of Appendix C. We restricted our experiments to English models commonly used and benchmarked on general reasoning datasets. Our datasets include those which are widely used in CoT and reasoning literature, including a mix of non-symbolic, semisymbolic, and symbolic reasoning. They span different formats, including multiple-choice, short-answer, and free-response; however, most of these datasets are multiple choice or short answer, as CoT is not typically used in long-form response settings. We also categorize each dataset into a larger category of reasoning required to solve it: Commonsense, Knowledge, Symbolic, Mathematical, and Soft Reasoning. We define Soft Reasoning as questions relying on commonsense and natural language but going beyond simple inferences about these statements. Finally, we explore several prompting strategies for eliciting reasoning from language models, as past work has emphasized the importance of the prompt (Yang et al., 2024). However, we generally found slight performance differences; see Appendix D for details. We therefore focus on prompts similar to Kojima et al. (2022) and Wei et al.

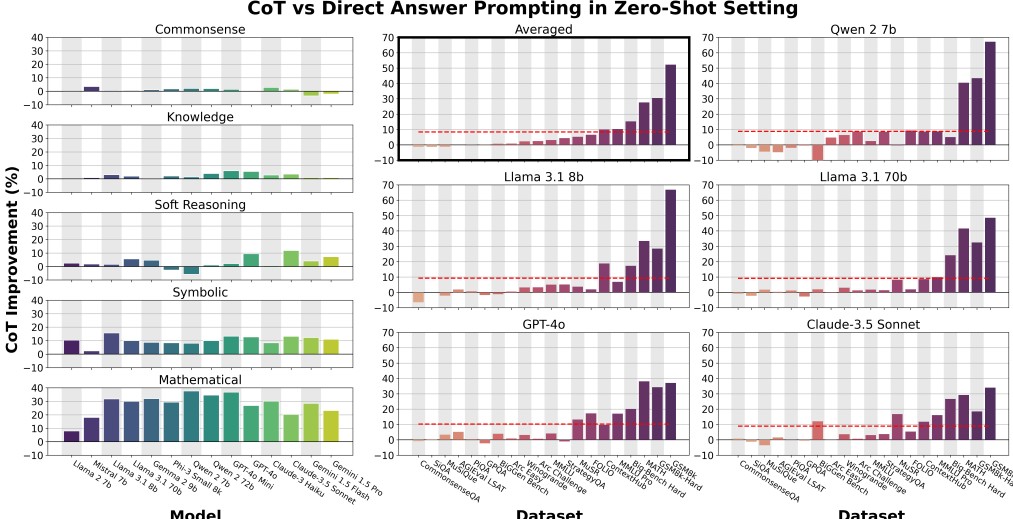

Figure 3: Left: Performance gain from using CoT for each reasoning category. Right: Performance gain from using CoT for each dataset, averaged across models and broken out across 5 representative models. Red lines indicate median improvement. In both plots we see a consistent trend: most improvements from using CoT are from math and symbolic reasoning.

(2022) for zero-shot and few-shot settings, respectively, with alterations to improve the model's ability to produce desired behavior (i.e., formats that allow for easily parsed answers). We upload all our prompts and outputs for each model for each prompting strategy on Huggingface.[4]

**Implementation Details**  We use a high-throughput inference package, vLLM (Kwon et al., 2023), for the model inference process. We use greedy decoding on all models. Our prompts are taken from the Llama 3.1 evaluations when available (Dubey et al., 2024), and minor adjustments are made to unify prompting strategies. For other datasets, we either use the standard prompt for the dataset from the corresponding original paper or implement our own prompt. Our answer parser ($\mathrm{extract}$) is tailored to each dataset and model. Specific details about each dataset, its prompts, and answer extractor can be found in Appendix C.

## 4.2 Results

**Where does zero-shot CoT improve over direct prompts?**  *On datasets that require math (MATH, GSM8K) or formal logic (ContextHub, MuSR to a lesser degree) to answer the problem.*

Figure 3 on the left shows the average CoT performance improvement for each reasoning category from Figure 1 (right); raw numbers can be found in Table 6 of the Appendix. On the right, Figure 3 shows the performance gain from using CoT for each dataset, averaged across all models and for a selection of individual models. On non-symbolic reasoning categories and datasets, specifically those that contain questions primarily involving commonsense (CSQA, PIQA, SiQA), language understanding (WinoGrande), and reading comprehension (AGI LSAT, ARC-Easy, ARC-Challenge), there is little to no separation between the performance of zero-shot CoT and zero-shot direct answer. Despite these datasets involving reasoning, CoT does not yield improvement.

By contrast, the mathematical and symbolic categories get larger boosts in improvements alongside symbolic and many semi-symbolic datasets. MATH and GSM8K show gains as large as 41.6% and 66.9%, respectively. The semi-symbolic datasets like ContextHub and MuSR Murder Mysteries show moderate gains. These datasets require the application of logical rules to reach the answer, e.g., first-order logic parsed from simple natural language (ContextHub) or more complex commonsense statements (MuSR Murder Mysteries). All results are shown in the Appendix F.1 as well as a full list of numeric results for both CoT and direct answer prompting in Table 7. We also explored the few-shot setting and found it had little impact on when CoT will help; see Appendix E.

---

[4]https://huggingface.co/collections/TAUR-Lab/cot-analysis-project-66bbb9e5e0156e65059895f5

**Does the answer format impact where CoT will help?**   *Not much. Free response capabilities required for BigGen Bench may not benefit from pre-planning.*

Many of the commonly-used datasets for problems other than math are multiple choice. We highlight here that CoT has similar performance to direct answer across models for two datasets that are not multiple-choice and contain varying levels of non-symbolic reasoning. First, MuSiQue (Trivedi et al., 2022) is a short-form QA task requiring multi-hop reasoning. We consider this a semi-symbolic dataset as the questions have an explicit multi-hop structure. Because answer spans in MuSiQue can be paraphrased in many different ways, we use GPT-4o to judge if two answer spans are equivalent. Despite being semi-symbolic, we see no overall improvement from CoT.

Second, BiGGen Bench (Kim et al., 2024) uses free-form responses as the answer to a question, and an LLM-as-a-judge is used to evaluate these responses on a scale of 1 to 5. Because free-form responses blur the lines between CoT and direct answering, we create a new prompt that asks the language model to plan the free response before giving it. We then only pass the free response to the judge (GPT-4o-mini in our case) with the prompt from Kim et al. (2024). We also filter out any questions that explicitly state "Think step-by-step". We plot the performance of BiGGen Bench as the number of times a response receives a score of 4 or better. Despite including many reasoning questions (including several categories of math) and other categories, such as planning, we only see a mild improvement here. Because previous experiments show CoT helping on similar types of questions in the QA format, the lack of similar improvements here could imply that pre-planning is insufficient for unlocking reasoning capabilities in the LLM. Future work is needed to prove this.

**Are the gains in Knowledge, Soft Reasoning, and Commonsense significant?**   *Mostly no, except for MMLU, StrategyQA, and MuSR.*

We tested the significance of the improvements from CoT on the 13 datasets in the Knowledge, Soft Reasoning, and Commonsense reasoning categories using paired bootstrapping to assess whether CoT gives a significant improvement. To account for multiple comparisons, we applied a Bonferroni correction, setting the p-value to 0.00027 to account for the 14 models and 13 datasets. About 32% (59) of the datasets that show a benefit in these three reasoning categories were considered significant. Nearly half of these comparisons (26) are on MMLU and MMLU Pro. On these datasets, we find that CoT is mainly helping on math-related questions. StrategyQA and MuSR also received a consistent performance boost across 10 and 6 models respectively. StrategyQA is often used to benchmark reasoning methods and is built specifically to get a benefit from methods that decompose the question into steps, so a gain in performance is not unprecedented. MuSR, similarly, was built to have multiple steps of complex natural language reasoning, which may receive benefits from CoT. The remaining datasets that receive significant benefits are spread across the datasets and models.

**Why do MMLU and MMLU Pro get a boost?**   MMLU and MMLU Pro contain many different questions requiring different types of reasoning. We separated MMLU and MMLU Pro questions into two bins, those related to math and those not related to math, by checking if the questions text or generated response from the LLM includes an "=". Figure 4 shows that a majority of the performance gain seen from MMLU and MMLU Pro is from the math slices of each dataset. See more details in Appendix G.

## 5   STRENGTHS AND WEAKNESSES OF COT AT FORMAL REASONING

Previous sections establish that CoT primarily helps with symbolic reasoning tasks, but not why. Many symbolic and semi-symbolic tasks be broken down into two stages (Ye et al., 2023; Pan et al., 2023; Jiang et al., 2024): planning, either via a formal or informal specification via prompting (Sun et al., 2024; Wang et al., 2023b), and execution, using the same LM or external solvers. In this section, we attribute the performance gains from CoT on symbolic tasks to these two stages.

Given a question that requires symbolic reasoning, we define the **planning** stage as extracting all variables from the context into a formal specification and defining their relations. The **execution** stage uses a solver that takes as input a plan and can be run in an orderly fashion to derive the final answer. Using our notation from Section 2, let $f(\mathbf{q}) = \mathcal{I}_{\text{planning}}^m(\mathbf{q})$ be a mapping of the question $\mathbf{q}$ to a symbolic plan $\mathcal{S}_{\text{plan}}$ that can be executed by the language model or by an external symbolic solver, $\hat{a} = \text{solve}(\mathcal{S}_{\text{plan}})$, where $\hat{a}$ is the final answer for $\mathbf{q}$.

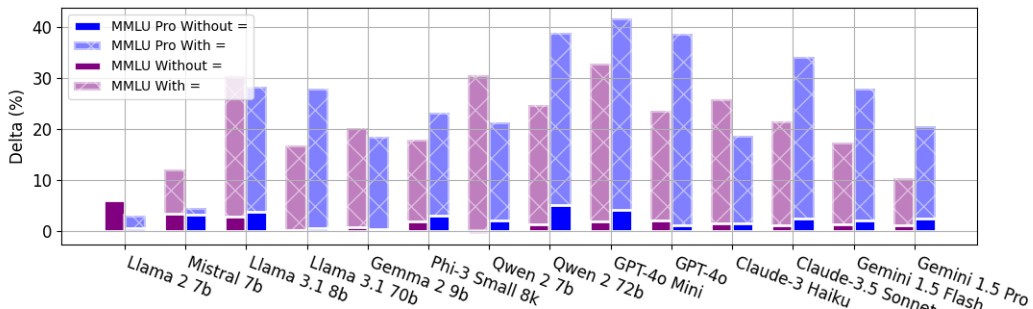

Figure 4: CoT deltas between MMLU and MMLU Pro performance when a question or generated response contains an "=" (With =) or not (Without =). We filter out any questions that do not result in a final answer (degeneration, etc.). CoT primarily helps on the pairs of questions and generations that contain an "=", which indicates math-related questions.

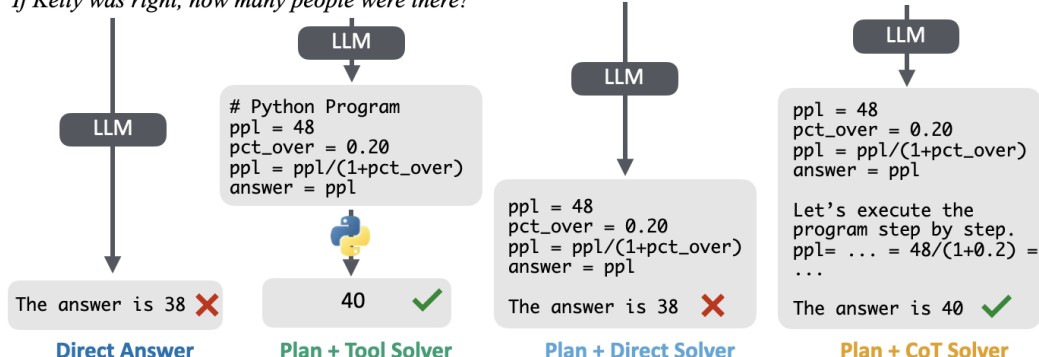

Figure 5: Prompt variants that separate planning and execution for GSM8K. For all prompt variants besides direct answer and CoT (not shown), we few-shot prompt an LLM to first generate a Python program as a solution plan. For Plan + Direct Solver, the LLM is prompted to directly give an answer from the plan; for Plan + CoT Solver, the LLM is prompted to solve the plan step-by-step with CoT and give an answer; for Plan + Tool Solver, we feed the plan into a Python interpreter.

By separating planning and execution in this way, we can test how much a language model can gain from only having a plan, to having a plan and solving it with CoT, or to having a plan and then solving it with an external symbolic solver. Given a plan $\mathcal{S}_{\mathrm{plan}} \sim \mathcal{I}_{\mathrm{planning}}^{m}(\mathbf{q})$, we compare the performance of the settings below to evaluate at which stage LM is most effective and falls short.

## 5.1 SETTINGS EVALUATED

**Settings 1 and 2: Few-shot direct answer and CoT:** We use the few-shot direct answer and CoT prompts from Section 4.1 as baselines. Figure 5 includes an example of each setting on GSM8K.

**Settings 3 and 4: Plan + Direct Solver and Plan + CoT Solver:** Here we use inspiration from Xu et al. (2024a) and generate a symbolic plan using the same strategy as Ye et al. (2023). Specifically, we use a few-shot prompt $\mathcal{I}_{\mathrm{planning}}^{m}$ to generate a formal specification $\mathcal{S}_{\mathrm{plan}}$ that should be executable by a symbolic solver. In the same prompt LMs are asked to solve their generated specification $\mathcal{S}_{\mathrm{plan}}$ and derive the final answer $\tilde{\mathbf{y}} \sim p(\mathbf{y} \mid \mathcal{I}_{\mathrm{da}}(\mathcal{S}_{\mathrm{plan}}))$, either directly giving the answer after generating the specification (**Plan + Direct Solver**) or providing step-by-step explanations and tracking of intermediate steps for the derivation (**Plan + CoT Solver**). Particularly, $\mathcal{S}_{\mathrm{plan}}$ is a Python program for math datasets, and is a set of first-order logic specifications for logical reasoning datasets.

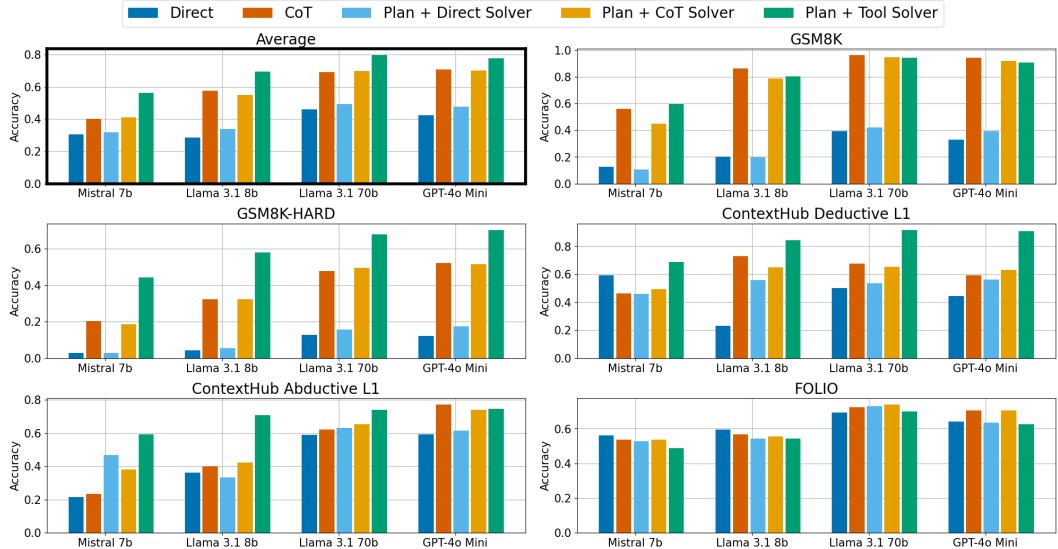

Figure 6: Performance of prompt variants that separate planning and execution for math and logical reasoning datasets. Despite outperforming direct answer for solving a formal plan and deriving the final answer, CoT is still limited in performing symbolic computations: there is a large performance boost from Plan + Tool Solver over CoT and Plan + CoT Solver on average across all models.

**Setting 5: Plan + Tool Solver:** We then evaluate how effective CoT can be at performing symbolic computations compared with external symbolic solvers. Following prior work on augmenting LMs with tools for math and logic questions (Ye et al., 2023; Pan et al., 2023; Gao et al., 2023; Chen et al., 2023), we generate $\mathcal{S}_{\text{plan}}$ the same way as in CoT Solver, but now feed in the plan into a symbolic solver (Python interpreter or a SMT Solver), such that $\hat{a} = \text{solve}(\mathcal{S}_{\text{plan}})$.

**Evaluation Setup:** We compare the performance of each setting on math (GSM8K) and logical reasoning (ContextHub and FOLIO) datasets. We follow Gao et al. (2023) to include GSM8K-Hard, a minimally modified version that replaces numbers of GSM8K with larger numbers, to account for the possibility of recent LLMs overfitting GSM8K by data contamination (Zhang et al., 2024).

For Plan + Direct solver and Plan + CoT solver, we use the few-shot prompts from Ye et al. (2023). For Plan + Tool solver, we use state-of-the-art tool-augmented prompting methods. Particularly, for GSM8K, we use Program-aided Language Model (Gao et al., 2023, PAL) that executes the LM-generated plan with a Python interpreter. For logical reasoning datasets, we use Satisfiability-Aided Language Model (Ye et al., 2023, SatLM) that uses automated theorem prover Z3 (De Moura & Bjørner, 2008) to solve the generated specifications. If the generated plan cannot be parsed by the tool, we use random guessing when the question is multiple choice, and mark it incorrect otherwise.

## 5.2 EVALUATION RESULTS

Figure 6 shows the results across a representative selection of models. Detailed numerical results, including the unparseable rates of model-generated plans, can be found in Appendix H.

When comparing direct answer with Plan + Direct solver and Plan + CoT solver, we note that for many datasets and models, only having a plan does not account for most of the performance gain. **Compared with direct answer, CoT or Plan + CoT solver is needed for strong performance. Tracking the execution with one of these methods gives the strongest accuracy benefit, especially for math-heavy datasets.**

Despite their strength over direct answer and Plan + Direct solver, **CoT and Plan + CoT solver are dominated by Plan + Tool solver in most settings**. LLMs are limited by their ability to execute and track steps compared with symbolic solvers.

We argue that these results provide an explanation of why CoT helps on symbolic tasks. While all tasks could feasibly benefit from a detailed description of how to solve each individual question (e.g., a *plan* in the context of this section), CoT only outperforms direct answer when these steps require a substantial amount of tracing and computation. In these settings, we can see clear performance

benefit from using symbolic solvers; **CoT appears to be a poor (but universal) approximation to such solvers**. When possible, LLMs should be paired with symbolic solvers at inference time when solving symbolic tasks to achieve consistently better performance over direct answer **and** CoT.

## 6 DISCUSSION AND RELATED WORK

**Where is CoT helping and why?**   Our results showing CoT improvement for math and logic aligns well with early work on CoT for LLMs such as Scratchpads (Nye et al., 2022). As CoT gained popularity, its application has broadened to tasks that canonically do not require multiple steps. It can often yield small improvements over direct answering. We believe this led to the current prevailing sentiment that deliberation should improve performance on any task requiring some type of reasoning (our original claim from Section 2). However, our results show a clear separation between performance on non-symbolic and symbolic tasks. If, in theory, any question could benefit from deliberation, why is CoT only benefiting the questions that can be solved through symbolic manipulation? Our results from Section 5 suggest that the primary benefit of CoT comes in the ability to execute symbolic steps and track their output. Not all tasks have this feature: for example, questions from CommonsenseQA can hardly be translated into formally grounded and executable solution plans. Datasets like StrategyQA may feature multiple steps of reasoning, but executing those steps is not complex, so the benefits of CoT are small. It is unclear whether explicitly instilling models with particular modes of deliberation, like process of elimination for multiple choice questions, might make them more effective for non-symbolic tasks, or whether there's a fundamental limitation imposed by their pre-training data. We leave this distinction for future work.

**Can we improve CoT further?**   Our work treats chain-of-thought variants that explicitly don't involve multiple inferences. There is evidence that using additional calls to LLMs can help (Du et al., 2023; Yao et al., 2023; Besta et al., 2023; Chen et al., 2024), but these methods use significantly increased computation, and careful benchmarking sometimes reveals that naive techniques are as good as iterative ones (Olausson et al., 2024). However, past theoretical results show that Transformers are augmented in a fundamental way by CoT (Liu et al., 2023b; Merrill & Sabharwal, 2024); we believe this indicates the potential for improving CoT beyond prompt-based CoT. On the other hand, recent methods showing benefit from "internalizing" CoT (Deng et al., 2024) may indicate that explicit generation of intermediate tokens is not used to its full potential.

**Limitations**   One set of tasks we do not cover in our experiments (except for BiGGen Bench) is long-horizon planning. However, many works in the literature have already discussed the efficacy of planning with CoT. We also do not address the data contamination of some of these models on the datasets. We try to mitigate this by including multiple models, datasets (new and old), and our meta-analysis. For more discussion of planning and dataset contamination, see Appendix I.

## 7 CONCLUSION

In this work, we characterize the performance of prompt-based CoT through a meta-analysis of the literature and experiments across different models, datasets, and prompts. We find that CoT predominantly helps on math and formal logic, largely due to its ability to trace the intermediate steps of a problem. But CoT rarely outperforms tool-augmented approaches for these same problems. We believe that CoT remains a powerful technique, but to give improvement across a wider range of NLP tasks, research should move beyond prompt-based CoT to new paradigms like search, interacting agents, or better fine-tuned models.

## REPRODUCIBILITY

For our experiments, we provide in-depth details of how we evaluated models on each dataset in Section 4.1 and Appendix C. Furthermore, we release all prompts for every dataset on Huggingface, including per model output and sampling parameters. For our meta-analysis of the literature, we describe our filtering criteria and process of annotating experiments into high-level categories in Section 3 and Appendix B. We also release the full list of papers in our meta-analysis together with extracted experimental comparisons and task category annotations.

ACKNOWLEDGMENTS

We acknowledge George Tsoukalas for providing insightful feedback throughout the project. We also thank Kaj Bostrom and Eunsol Choi for reviewing and providing feedback on drafts of the work. This work was partially supported by NSF CAREER Award IIS-2145280 (to Durrett), NSF CAREER Award 2339729 (to Mahowald), the NSF AI Institute for Foundations of Machine Learning (IFML), the Sloan Foundation via a Sloan Research Fellowship, and a grant from Open Philanthropy.

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

## A  META-ANALYSIS EXPANDED DETAILS ON CRITERIA AND PROCESS

**Automatic Selection and Paper Filtering**   We investigate all papers from ICLR 2024, a representative ML venue, and two representative NLP venues, EACL 2024 and NAACL 2024 (including Findings and Workshop papers). We filtered all 4,642 papers (2,259 from ICLR 2024 and 2,382 from the two ACL-affiliated conferences) for those with at least two occurrences of "CoT", "chain-of-thought", or "chain of thought", resulting in 516 papers. There are conceivably papers using CoT called by another name (e.g., Scratchpads), but we believe these 516 give a representative sample appropriate for systematic analysis.

**Manual Paper Filtering and Results Extraction**   We then filter down to papers that perform a comparison of CoT prompting vs. direct prompting, whether or not this is core to the paper's research question. We manually filtered the 516 papers in question and extracted the key results from those that remained. We excluded multimodal models, CoT-fine-tuned models, any experiments where the "CoT" method involves multiple forward passes (e.g., self-consistency (Wang et al., 2023c) and tree-of-thought (Yao et al., 2023)),[5] and systems that augment LLMs with external tools (discussed more in Section 5).

For each paper passing through these criteria, we manually extracted the results from key tables comparing CoT and direct answer prompts. We only include results where the CoT and direct prompts are run on the same model and same dataset while being on a scale of 0 to 100 (excluding Likert scale evaluations, for example) for a more direct comparison. When papers include various CoT or direct answer prompts (including zero/few-shot variants), we always take the best-performing prompt for both. We focus on key test results where applicable, excluding dev sets if they are reported alongside test and also excluding numbers from ablations or nonstandard subsets of datasets.

This resulted in a total of 1,218 experimental comparisons across 110 papers (35 from ICLR and 75 from NAACL and EACL) covering 264 datasets. Details and more information will be available in our GitHub Repo.

**Categorization**   Given the large number of tasks and datasets being compared, we grouped each task into a set of 14 categories. These categories were determined based on the description (and possibly examples) of the task, not taking into account system performance. These categories abstract over traditional NLP task classifications (e.g., NER, reading comprehension) and take into account both the task format and the kinds of reasoning involved. Definitions for several categories are shown in Table 1 and the full description is given in Appendix B.

## B  QUANTITATIVE META-ANALYSIS

See the full list of categories and their descriptions that we used for the meta-analysis in Table 2.

## C  EXPANDED EXPERIMENTAL DETAILS

A full list of the datasets can be found in Table 4. Each model can be seen in Table 5. We use one answer parser for all datasets of the same answer response format (one for multiple choice, short answer, etc.); however, some datasets require special handling and have edge cases that we handle separately from the rest of the datasets. Similarly, for each model, we use the exact same prompt across them, except when closed source models require different prompts because they do not allow for partial completions (i.e., when we cannot put "*let's think step by step*" to warm-start the assistant's response). All prompts are given in our Huggingface repo, including the model output and what our answer parser extracted as the answer.

Experiments were conducted either by invoking APIs or by running open-source models on our own hardware, mostly on a machine with 8 A40s or 4 Quadro RTX 8000s. All locally hosted models were

---

[5]These systems use more compute than direct answer, and there is not a clear comparison to be made here. Moreover, our anecdotal coverage of these methods shows that they are most used for math, coding, and logic settings, for which we already have high representation among reported CoT methods.

Table 2: Categories and their descriptions for the meta-analysis.

| Category | Description |
|---|---|
| Symbolic and algorithmic | Tasks involving symbol manipulation which can be solved by executing a program. This includes entity tracking datasets (e.g., SCONE, Coin Flip) and algorithmic tasks (e.g., BBH word sorting or finding shortest paths in a graph). |
| Math | Tasks requiring mathematical reasoning, from grade-school math to advanced mathematics, including physics questions. |
| Logical reasoning | Tasks designed to test for logical reasoning, whether deductive (Saparov & He, 2023, PrOntoQA), inductive (Bowen et al., 2024) or analogical (Ma et al., 2024) reasoning, including syllogisms and logical puzzles. |
| Commonsense reasoning | Datasets designed to test for commonsense knowledge and reasoning, i.e., world knowledge that most people would have, rather than specialized expert-level knowledge in a discipline acquired after years of study. |
| Encyclopedic knowledge | Tasks requiring expert-level in-depth knowledge beyond mere commonsense, usually in an open-book setting. |
| Spatial and temporal reasoning | Datasets designed to test for an understanding of space and spatial relations (e.g., navigation) or reasoning involving time and sequences over time. |
| Multi-hop QA | Questions involving the composition of multiple steps of reasoning in order to arrive at an answer, such as "What is the capital of the country whose scientist discovered penicillin?" |
| Context-aware QA | Tasks such as closed-book QA and reading comprehension involving reasoning about a given text in context. The context is often a short passage, but could also take the form of a knowledge graph (KBQA) or a table. This category also includes information extraction tasks, such as NER or relation extraction. |
| Entailment | Tasks involving establishing the inferential relation between two texts, prototypically NLI, but also including fact verification. |
| Text classification | Tasks involving the classification of a text into a small set of categories, such a topic or sentiment classification, but also involving tasks such as hate speech detection and misinformation detection. |
| Generation | Tasks involving text generation, including machine translation, dialogue, question generation, as well as code generation. Tasks such as SQL execution (Lei et al., 2024) or systematic transformations of data (e.g., SCAN (Lake & Baroni, 2018)) are excluded because they can be solved by executing a program. |
| Meta-linguistic | Tasks probing for models' knowledge of linguistics, such as identifying the main subject of a sentence or solving linguistic puzzles. |
| Mixed datasets | Datasets containing a variety of tasks, such as BIG-Bench Hard (BBH) or MMLU. |
| Other | Tasks which did not fit in any of the other categories, such as evaluating AI safety, eliciting models' verbalized confidence, or melody retrieval. |

hosted with vLLM. All parameters given to the vLLM API endpoint are given in the Huggingface repo as well.

## D    OTHER CoT PROMPT VARIANTS

### D.1    TESTING PERFORMANCE VOLATILITY ACROSS PROMPTS

To test the impact of prompt choice on performance, we performed our zero-shot experiment on Llama 3.1 8B with 7 different datasets and 4 different zero-shot CoT prompting strategies common in the literature (Kojima et al., 2022; Wang et al., 2023b; Zhou et al., 2023b; Yang et al., 2024).

Table 3: Models, datasets, and prompting strategies used in our experiments. Models marked with †
are run with a 4k context size window. Note that Gemma has a larger than 4k context size window,
but VLLM only supports up to a 4k context size window for it. Models marked with * indicate
closed-source models that cannot handle prefixed assistant messages. Datasets marked with △ do
not have a few-shot setting.

| | |
|---|---|
| **Models** | Llama 2 7B Chat † (Touvron et al., 2023), Mistral 7B Instruct v0.3 (Jiang et al., 2023), Llama 3.1 8B Instruct (Dubey et al., 2024), Llama 3.1 70B Instruct, Gemma 2 9B It† (Riviere & et. al, 2024), Phi-3 Small 8k Instruct (Abdin et al., 2024), gpt-4o-mini-2024-07-18*, gpt-4o-2024-08-06*, Gemini 1.5 Flash*(Reid & et. al, 2024), Gemini 1.5 Pro*(Reid & et. al, 2024), claude-3-haiku-20240307* (Anthropic, a), claude-3-5-sonnet-20240620* (Anthropic, b) |
| **Datasets** | CommonsenseQA (Talmor et al., 2019), StrategyQA (Geva et al., 2021), SiQA△ Sap et al. (2019), PiQA△ (Bisk et al., 2019), Winogrande△ (Sakaguchi et al., 2021), GPQA (Rein et al., 2023), MuSR (Sprague et al., 2024), ContextHub (Levels 1 and 2 only) (Hua et al., 2024), ARC△ (Clark et al., 2018), AGIEval LSAT (Zhong et al., 2023), MMLU (Hendrycks et al., 2021a), MMLU Pro (Wang et al., 2024), MATH (Hendrycks et al., 2021b), GSM8K (Cobbe et al., 2021), GSM8K-hard (Gao et al., 2023), FOLIO (Han et al., 2022), MuSiQue△ (Trivedi et al., 2022), Big-Bench Hard (Suzgun et al., 2023; Srivastava et al., 2022), BiGGen Bench (Kim et al., 2024) |
| **Prompts** | zero-shot direct answer, zero-shot CoT (Kojima et al., 2022), few-shot direct answer (Brown et al., 2020), few-shot CoT (Wei et al., 2022) |

Table 4: List of datasets used in our experiments. We categorize each dataset into one of five
categories based on the type of reasoning required: Commonsense, Knowledge, Soft Reasoning,
Symbolic, or Mathematical. We also report answer formats. When we use few-shot prompts, we
mark how many examples those prompts contain. BiGGen Bench has many categories of questions
that explicitly ask for CoTs in the response; we ignore those categories for our evaluation.

| Dataset | Type | Answer Format | $m$-Shots |
|---|---|---|---|
| CommonsenseQA | Commonsense | Multiple choice | 7 |
| StrategyQA | Commonsense | True or False | 6 |
| SIQA | Commonsense | Multiple choice | 0 |
| PIQA | Commonsense | Multiple choice | 0 |
| Winogrande | Commonsense | Multiple choice | 0 |
| Arc Easy | Knowledge | Multiple choice | 0 |
| Arc Challenge | Knowledge | Multiple choice | 0 |
| AGIEval LSAT | Soft Reasoning | Multiple choice | 3 |
| BiGGen-Bench | Soft Reasoning | Free response | 0 |
| MMLU | Knowledge | Multiple Choice | 5 |
| MMLU Pro | Knowledge | Multiple Choice | 5 |
| BigBench-Hard | Symbolic | Multiple Choice | 0 |
| MuSR | Soft Reasoning | Multiple Choice | 1 |
| GPQA | Mathematical | Multiple Choice | 3 |
| MuSiQue | Soft Reasoning | Short Answer | 0 |
| GSM8K | Mathematical | Short Answer | 8 |
| GSM8K-Hard | Mathematical | Short Answer | 8 |
| FOLIO | Symbolic | True, False, or Unknown | 4 |
| ContextHub | Symbolic | True, False, or Neither | 3 |
| MATH | Mathematical | Short Answer | 4 |

Table 5: List of models for our experiments. We focus on contemporary instruction-tuned models; although pretrained and smaller language models could be used, they are not the focus of our study. Prompts and outputs used for each model are available on Huggingface. * Note that Gemma can accept more than 4k input tokens, but we are restricted to 4k by vLLM.

| Model | Context Length | Is Open Source |
|---|---|---|
| Llama 2 7B Chat | 4k | True |
| Mistral 7B Instruct v0.3 | 8k | True |
| Llama 3.1 8B Instruct | 128k | True |
| Llama 3.1 70B Instruct | 128k | True |
| Gemma 2 9B It | 4k* | True |
| Qwen 7B Instruct | 131k | True |
| Qwen 72B Instruct | 131k | True |
| GPT4o-Mini | 128k | False |
| GPT4o | 128k | False |
| Gemini 1.5 Pro | 128k | False |
| Gemini Flash | 1m | False |
| Claude 3.5 Sonnet | 200k | False |
| Claude 3 Haiku | 200k | False |

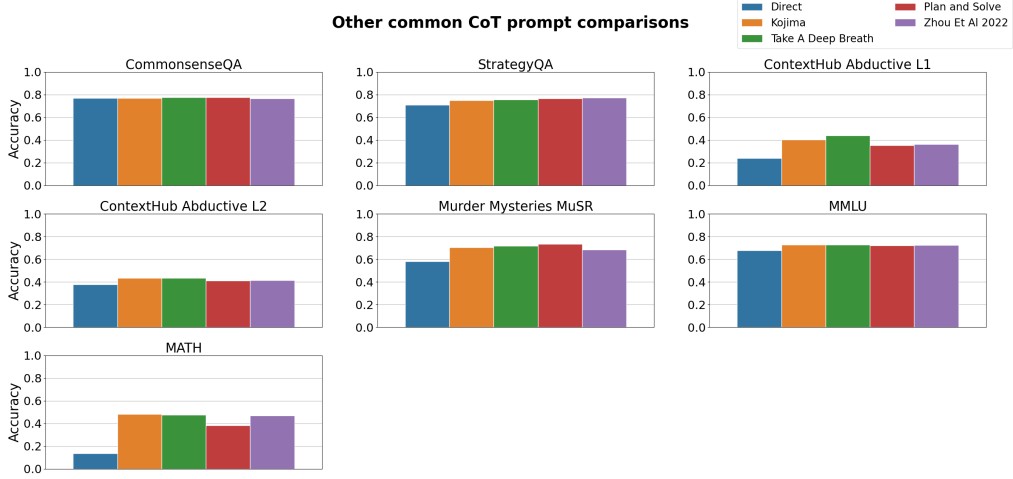

Figure 7: Performance of multiple prompts commonly used to elicit reasoning through CoT in the zero shot setting. Each prompt starts the assistant completion with a different phrase meant to elicit reasoning. All results are from using Llama 3.1 8B Instruct. For the Kojima variant, we explicitly place "*Let's think step by step.*" in the assistant message. There is very little variation between the CoT prompts on average.

Figure 7 shows variation due to prompts is typically small and no prompt gives a consistent gain over the other. For our experiments, this suggests that different prompts have small effects on the overall outcome on average.

## E  FEW-SHOT EXPERIMENTS

Compared to a zero-shot prompt, a few-shot prompt additionally contains demonstrations of the relevant reasoning mode on different problem instances $\{(v(\mathbf{q}_i), \mathbf{y}_i^*)\}$. Few-shot prompts for direct answer simply encode the answer $a_i$ as $\mathbf{y}_i^*$, whereas few-shot prompts for chain-of-thought include a reasoning trace ending in the correct answer. Now we can define the $m$-shot direct prompt as

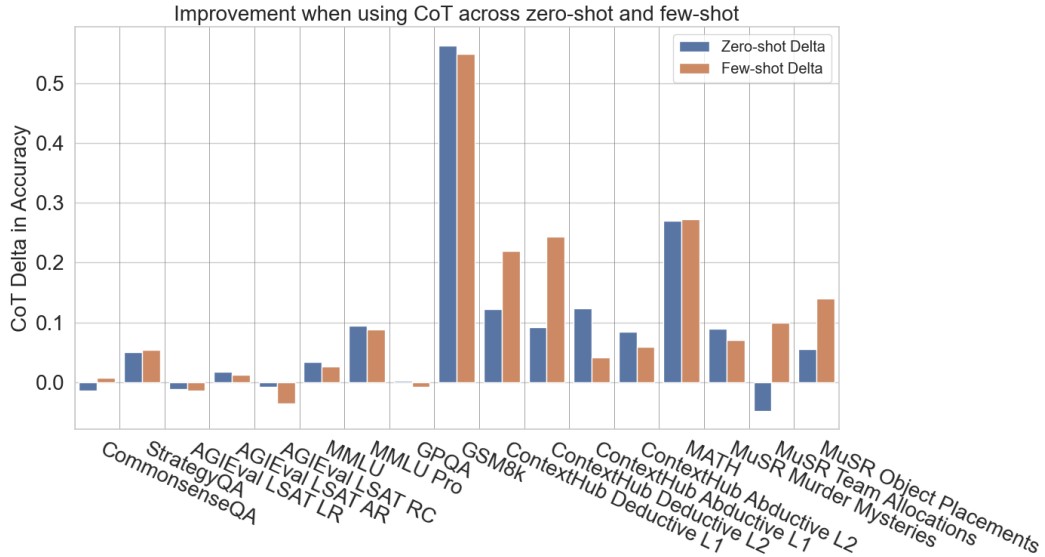

Figure 8: Average performance improvement from using CoT across different models in the zero-shot and few-shot settings. Each bar represents how much CoT improves the accuracy for that specific setting. In general, CoT in the few-shot setting does not change the qualitative performance of CoT versus zero-shot, though it can change the magnitude for symbolic datasets.

$\mathcal{I}_{\text{da}}^m(\mathbf{q}) = v_{\text{da}}(\mathbf{q}_1)\mathbf{a}_1 v_{\text{da}}(\mathbf{q}_2)\mathbf{a}_2 \ldots v_{\text{da}}(\mathbf{q}_m)\mathbf{a}_m v_{\text{da}}(\mathbf{q})$ and the $m$-shot cot prompt as $\mathcal{I}_{\text{cot}}^m(\mathbf{q}) = v_{\text{cot}}(\mathbf{q}_1)\mathbf{y}_1^* v_{\text{cot}}(\mathbf{q}_2)\mathbf{y}_2^* \ldots v_{\text{cot}}(\mathbf{q}_m)\mathbf{y}_m^* v_{\text{cot}}(\mathbf{q})$.

Figure 8 shows the difference between few-shot prompting and the zero-shot setting discussed in the main text of the paper. We see that using CoT in the few-shot setting largely does not change the datasets that benefit from it. Only one dataset, MuSR Team Allocation, starts to improve with few-shot; however, we believe this to be an exception because the final step to derive the answer is complex in the prompt and clearer in the examples. The magnitude of improvement over direct answer prompting when using CoT is also similar to the zero-shot setting.

# F  EXPANDED COT VS DIRECT EXPERIMENTAL RESULTS

## F.1  FULL ZERO-SHOT RESULTS

Table 6: Direct answer and CoT accuracies for each reasoning category across models.

| Model | Commonsense | | Knowledge | | Mathematical | | Symbolic | | Soft | |
|---|---|---|---|---|---|---|---|---|---|---|
| | DA % | CoT % | DA % | CoT % | DA % | CoT % | DA % | CoT % | DA % | CoT % |
| Claude-3 Haiku | 74.3 | 77.2 | 73.0 | 76.1 | 18.1 | 48.2 | 38.6 | 48.7 | 55.9 | 56.6 |
| Claude-3.5 Sonnet | 84.3 | 85.8 | **83.8** | **88.8** | **38.7** | 59.0 | 53.2 | 67.1 | **67.6** | **75.7** |
| GPT-4o Mini | 81.8 | 83.2 | 73.6 | 83.1 | 22.9 | 59.7 | 48.1 | 60.9 | 61.1 | 63.5 |
| Gemini 1.5 Flash | 80.3 | 76.8 | 78.2 | 81.0 | 27.2 | 55.7 | 47.0 | 59.7 | 60.6 | 62.6 |
| Gemini 1.5 Pro | 80.4 | 78.3 | 80.9 | 83.8 | 35.4 | 58.5 | 52.9 | 62.6 | 64.1 | 67.8 |
| Gemma 2 9b | 75.0 | 76.1 | 74.9 | 76.9 | 18.5 | 50.5 | 46.7 | 55.8 | 58.2 | 60.5 |
| Gpt-4o | **87.3** | **87.7** | 82.9 | 88.6 | 36.5 | **63.3** | 55.7 | **68.3** | 65.9 | 74.0 |
| Meta-Llama 2 7b | 51.4 | 50.9 | 44.1 | 46.6 | 9.3 | 17.2 | 22.4 | 35.4 | 37.2 | 37.6 |
| Meta-Llama 3.1 70b | 84.2 | 84.7 | 82.4 | 85.6 | 24.9 | 54.9 | 49.0 | 60.0 | 65.7 | 69.5 |
| Meta-Llama 3.1 8b | 72.9 | 73.4 | 70.1 | 74.1 | 16.0 | 47.8 | 34.8 | 51.6 | 55.0 | 56.2 |
| Mistral 7b | 58.3 | 61.8 | 62.0 | 64.5 | 10.9 | 28.9 | 41.8 | 45.0 | 48.6 | 49.7 |
| Phi-3 Small 8k | 70.8 | 72.5 | 76.1 | 79.7 | 17.8 | 47.1 | 51.2 | 58.7 | 57.9 | 56.4 |
| Qwen 2 72b | 82.9 | 84.9 | 78.6 | 84.6 | 23.9 | 58.5 | 48.2 | 58.7 | 64.2 | 65.1 |
| Qwen 2 7b | 64.0 | 66.1 | 65.2 | 71.3 | 15.9 | 53.5 | 43.8 | 52.3 | 54.4 | 49.4 |
| Average | 74.8 | 75.7 | 73.3 | 77.5 | 22.6 | 50.2 | 45.2 | 56.1 | 58.3 | 60.3 |

Table 7: Zero-shot accuracy for direct answering and CoT prompts on all datasets

| Dataset | Type | Model | zero-shot CoT accuracy | zero-shot DA accuracy |
|---|---|---|---|---|
| MuSR Team Allocations | Soft Reasoning | Llama 2 7b | 34.8 | 37.2 |
| MuSR Team Allocations | Soft Reasoning | Mistral 7b | 38.8 | 46.8 |
| MuSR Team Allocations | Soft Reasoning | Llama 3.1 8b | 44.0 | 48.0 |
| MuSR Team Allocations | Soft Reasoning | Llama 3.1 70b | 65.2 | 66.8 |
| MuSR Team Allocations | Soft Reasoning | Gemma 2 9b | 47.2 | 44.8 |
| MuSR Team Allocations | Soft Reasoning | Phi-3 Small 8k | 47.2 | 61.6 |
| MuSR Team Allocations | Soft Reasoning | Qwen 2 7b | 42.0 | 49.6 |
| MuSR Team Allocations | Soft Reasoning | Qwen 2 72b | 58.0 | 66.8 |
| MuSR Team Allocations | Soft Reasoning | GPT-4o Mini | 61.2 | 58.4 |
| MuSR Team Allocations | Soft Reasoning | Gpt-4o | 64.0 | 63.6 |
| MuSR Team Allocations | Soft Reasoning | Claude-3 Haiku | 56.8 | 59.2 |
| MuSR Team Allocations | Soft Reasoning | Claude-3.5 Sonnet | 80.4 | 63.2 |
| MuSR Team Allocations | Soft Reasoning | Gemini 1.5 Flash | 48.8 | 55.2 |
| MuSR Team Allocations | Soft Reasoning | Gemini 1.5 Pro | 58.4 | 62.4 |
| SiQA | Commonsense | Llama 2 7b | 53.4 | 55.9 |
| SiQA | Commonsense | Mistral 7b | 35.9 | 33.5 |
| SiQA | Commonsense | Llama 3.1 8b | 73.5 | 73.5 |
| SiQA | Commonsense | Llama 3.1 70b | 78.7 | 80.9 |
| SiQA | Commonsense | Gemma 2 9b | 74.9 | 76.3 |
| SiQA | Commonsense | Phi-3 Small 8k | 38.0 | 40.4 |
| SiQA | Commonsense | Qwen 2 7b | 37.3 | 39.3 |
| SiQA | Commonsense | Qwen 2 72b | 80.5 | 80.4 |
| SiQA | Commonsense | GPT-4o Mini | 79.0 | 80.0 |
| SiQA | Commonsense | Gpt-4o | 81.9 | 81.5 |
| SiQA | Commonsense | Claude-3 Haiku | 75.4 | 74.8 |
| SiQA | Commonsense | Claude-3.5 Sonnet | 79.7 | 81.0 |
| SiQA | Commonsense | Gemini 1.5 Flash | 74.5 | 79.1 |
| SiQA | Commonsense | Gemini 1.5 Pro | 73.9 | 78.2 |
| MuSiQue | Soft Reasoning | Llama 2 7b | 40.1 | 36.1 |
| MuSiQue | Soft Reasoning | Mistral 7b | 47.3 | 47.2 |
| MuSiQue | Soft Reasoning | Llama 3.1 8b | 62.6 | 64.7 |
| MuSiQue | Soft Reasoning | Llama 3.1 70b | 74.0 | 72.2 |
| MuSiQue | Soft Reasoning | Gemma 2 9b | 67.7 | 68.7 |
| MuSiQue | Soft Reasoning | Phi-3 Small 8k | 58.3 | 64.3 |
| MuSiQue | Soft Reasoning | Qwen 2 7b | 60.7 | 65.1 |
| MuSiQue | Soft Reasoning | Qwen 2 72b | 56.3 | 69.0 |
| MuSiQue | Soft Reasoning | GPT-4o Mini | 71.3 | 68.2 |
| MuSiQue | Soft Reasoning | Gpt-4o | 73.5 | 70.1 |
| MuSiQue | Soft Reasoning | Claude-3 Haiku | 54.8 | 56.0 |
| MuSiQue | Soft Reasoning | Claude-3.5 Sonnet | 66.9 | 70.4 |
| MuSiQue | Soft Reasoning | Gemini 1.5 Flash | 69.8 | 66.2 |
| MuSiQue | Soft Reasoning | Gemini 1.5 Pro | 69.8 | 71.3 |
| AGIEval LSAT RC | Soft Reasoning | Llama 2 7b | 31.2 | 36.4 |
| AGIEval LSAT RC | Soft Reasoning | Mistral 7b | 61.7 | 61.0 |
| AGIEval LSAT RC | Soft Reasoning | Llama 3.1 8b | 71.0 | 68.8 |
| AGIEval LSAT RC | Soft Reasoning | Llama 3.1 70b | 84.4 | 87.0 |
| AGIEval LSAT RC | Soft Reasoning | Gemma 2 9b | 75.1 | 78.1 |
| AGIEval LSAT RC | Soft Reasoning | Phi-3 Small 8k | 68.8 | 69.9 |
| AGIEval LSAT RC | Soft Reasoning | Qwen 2 7b | 61.0 | 66.5 |
| AGIEval LSAT RC | Soft Reasoning | Qwen 2 72b | 83.6 | 84.4 |

Table 7: Zero-shot accuracy for direct answering and CoT prompts on all datasets

| Dataset | Type | Model | zero-shot CoT accuracy | zero-shot DA accuracy |
|---|---|---|---|---|
| AGIEval LSAT RC | Soft Reasoning | GPT-4o Mini | 77.3 | 74.3 |
| AGIEval LSAT RC | Soft Reasoning | Gpt-4o | 88.1 | 81.4 |
| AGIEval LSAT RC | Soft Reasoning | Claude-3 Haiku | 71.7 | 65.1 |
| AGIEval LSAT RC | Soft Reasoning | Claude-3.5 Sonnet | 90.0 | 89.6 |
| AGIEval LSAT RC | Soft Reasoning | Gemini 1.5 Flash | 78.1 | 81.0 |
| AGIEval LSAT RC | Soft Reasoning | Gemini 1.5 Pro | 82.2 | 85.9 |
| CommonsenseQA | Commonsense | Llama 2 7b | 49.4 | 54.6 |
| CommonsenseQA | Commonsense | Mistral 7b | 68.0 | 68.0 |
| CommonsenseQA | Commonsense | Llama 3.1 8b | 68.5 | 74.9 |
| CommonsenseQA | Commonsense | Llama 3.1 70b | 83.5 | 84.4 |
| CommonsenseQA | Commonsense | Gemma 2 9b | 79.2 | 80.1 |
| CommonsenseQA | Commonsense | Phi-3 Small 8k | 81.8 | 80.3 |
| CommonsenseQA | Commonsense | Qwen 2 7b | 78.5 | 79.0 |
| CommonsenseQA | Commonsense | Qwen 2 72b | 87.4 | 87.3 |
| CommonsenseQA | Commonsense | GPT-4o Mini | 82.5 | 83.9 |
| CommonsenseQA | Commonsense | Gpt-4o | 86.5 | 87.3 |
| CommonsenseQA | Commonsense | Claude-3 Haiku | 80.6 | 79.0 |
| CommonsenseQA | Commonsense | Claude-3.5 Sonnet | 85.1 | 84.3 |
| CommonsenseQA | Commonsense | Gemini 1.5 Flash | 79.7 | 82.6 |
| CommonsenseQA | Commonsense | Gemini 1.5 Pro | 79.9 | 82.9 |
| GPQA | Mathematical | Llama 2 7b | 28.3 | 24.3 |
| GPQA | Mathematical | Mistral 7b | 23.0 | 24.3 |
| GPQA | Mathematical | Llama 3.1 8b | 24.1 | 25.9 |
| GPQA | Mathematical | Llama 3.1 70b | 23.2 | 25.9 |
| GPQA | Mathematical | Gemma 2 9b | 26.3 | 21.2 |
| GPQA | Mathematical | Phi-3 Small 8k | 22.3 | 20.8 |
| GPQA | Mathematical | Qwen 2 7b | 24.1 | 24.6 |
| GPQA | Mathematical | Qwen 2 72b | 21.0 | 18.1 |
| GPQA | Mathematical | GPT-4o Mini | 21.0 | 24.0 |
| GPQA | Mathematical | Gpt-4o | 23.7 | 25.9 |
| GPQA | Mathematical | Claude-3 Haiku | 25.4 | 22.3 |
| GPQA | Mathematical | Claude-3.5 Sonnet | 25.4 | 25.9 |
| GPQA | Mathematical | Gemini 1.5 Flash | 22.3 | 22.8 |
| GPQA | Mathematical | Gemini 1.5 Pro | 21.0 | 23.7 |
| AGIEval LSAT LR | Soft Reasoning | Llama 2 7b | 29.4 | 33.5 |
| AGIEval LSAT LR | Soft Reasoning | Mistral 7b | 44.1 | 47.8 |
| AGIEval LSAT LR | Soft Reasoning | Llama 3.1 8b | 59.0 | 53.9 |
| AGIEval LSAT LR | Soft Reasoning | Llama 3.1 70b | 81.4 | 81.0 |
| AGIEval LSAT LR | Soft Reasoning | Gemma 2 9b | 64.9 | 67.6 |
| AGIEval LSAT LR | Soft Reasoning | Phi-3 Small 8k | 64.5 | 64.1 |
| AGIEval LSAT LR | Soft Reasoning | Qwen 2 7b | 50.6 | 58.4 |
| AGIEval LSAT LR | Soft Reasoning | Qwen 2 72b | 77.3 | 75.1 |
| AGIEval LSAT LR | Soft Reasoning | GPT-4o Mini | 65.3 | 68.2 |
| AGIEval LSAT LR | Soft Reasoning | Gpt-4o | 87.3 | 83.9 |
| AGIEval LSAT LR | Soft Reasoning | Claude-3 Haiku | 55.7 | 54.7 |
| AGIEval LSAT LR | Soft Reasoning | Claude-3.5 Sonnet | 83.7 | 82.7 |
| AGIEval LSAT LR | Soft Reasoning | Gemini 1.5 Flash | 70.0 | 71.2 |
| AGIEval LSAT LR | Soft Reasoning | Gemini 1.5 Pro | 79.4 | 80.4 |
| PiQA | Commonsense | Llama 2 7b | 62.1 | 64.7 |
| PiQA | Commonsense | Mistral 7b | 78.6 | 77.7 |
| PiQA | Commonsense | Llama 3.1 8b | 85.0 | 84.2 |
| PiQA | Commonsense | Llama 3.1 70b | 91.8 | 90.6 |
| PiQA | Commonsense | Gemma 2 9b | 84.0 | 84.8 |
| PiQA | Commonsense | Phi-3 Small 8k | 89.1 | 85.5 |
| PiQA | Commonsense | Qwen 2 7b | 84.3 | 86.2 |
| PiQA | Commonsense | Qwen 2 72b | 92.9 | 89.1 |
| PiQA | Commonsense | GPT-4o Mini | 93.1 | 88.6 |
| PiQA | Commonsense | Gpt-4o | 95.9 | 95.5 |
| PiQA | Commonsense | Claude-3 Haiku | 85.9 | 86.6 |
| PiQA | Commonsense | Claude-3.5 Sonnet | 94.6 | 94.5 |
| PiQA | Commonsense | Gemini 1.5 Flash | 84.6 | 89.8 |
| PiQA | Commonsense | Gemini 1.5 Pro | 88.1 | 91.3 |
| Arc Easy | Knowledge | Llama 2 7b | 71.1 | 69.8 |
| Arc Easy | Knowledge | Mistral 7b | 87.5 | 86.7 |
| Arc Easy | Knowledge | Llama 3.1 8b | 93.0 | 92.5 |
| Arc Easy | Knowledge | Llama 3.1 70b | 97.5 | 97.9 |
| Arc Easy | Knowledge | Gemma 2 9b | 94.9 | 95.8 |
| Arc Easy | Knowledge | Phi-3 Small 8k | 96.0 | 96.3 |
| Arc Easy | Knowledge | Qwen 2 7b | 89.5 | 84.7 |
| Arc Easy | Knowledge | Qwen 2 72b | 97.9 | 97.4 |
| Arc Easy | Knowledge | GPT-4o Mini | 96.8 | 94.6 |
| Arc Easy | Knowledge | Gpt-4o | 98.9 | 98.1 |
| Arc Easy | Knowledge | Claude-3 Haiku | 95.1 | 95.4 |
| Arc Easy | Knowledge | Claude-3.5 Sonnet | 98.6 | 98.4 |
| Arc Easy | Knowledge | Gemini 1.5 Flash | 96.8 | 97.2 |
| Arc Easy | Knowledge | Gemini 1.5 Pro | 97.2 | 94.6 |
| Arc Challenge | Knowledge | Llama 2 7b | 49.2 | 45.2 |

Table 7: Zero-shot accuracy for direct answering and CoT prompts on all datasets

| Dataset | Type | Model | zero-shot CoT accuracy | zero-shot DA accuracy |
|---|---|---|---|---|
| Arc Challenge | Knowledge | Mistral 7b | 78.3 | 76.6 |
| Arc Challenge | Knowledge | Llama 3.1 8b | 86.0 | 82.6 |
| Arc Challenge | Knowledge | Llama 3.1 70b | 95.0 | 93.6 |
| Arc Challenge | Knowledge | Gemma 2 9b | 91.0 | 89.6 |
| Arc Challenge | Knowledge | Phi-3 Small 8k | 91.6 | 91.0 |
| Arc Challenge | Knowledge | Qwen 2 7b | 83.9 | 75.3 |
| Arc Challenge | Knowledge | Qwen 2 72b | 96.3 | 94.6 |
| Arc Challenge | Knowledge | GPT-4o Mini | 93.3 | 82.6 |
| Arc Challenge | Knowledge | Gpt-4o | 96.0 | 95.3 |
| Arc Challenge | Knowledge | Claude-3 Haiku | 89.3 | 89.3 |
| Arc Challenge | Knowledge | Claude-3.5 Sonnet | 96.0 | 95.3 |
| Arc Challenge | Knowledge | Gemini 1.5 Flash | 92.3 | 93.6 |
| Arc Challenge | Knowledge | Gemini 1.5 Pro | 91.6 | 90.6 |
| AGIEval LSAT AR | Soft Reasoning | Llama 2 7b | 17.0 | 17.4 |
| AGIEval LSAT AR | Soft Reasoning | Mistral 7b | 21.7 | 19.1 |
| AGIEval LSAT AR | Soft Reasoning | Llama 3.1 8b | 20.4 | 26.1 |
| AGIEval LSAT AR | Soft Reasoning | Llama 3.1 70b | 32.6 | 28.7 |
| AGIEval LSAT AR | Soft Reasoning | Gemma 2 9b | 24.8 | 23.0 |
| AGIEval LSAT AR | Soft Reasoning | Phi-3 Small 8k | 28.3 | 26.5 |
| AGIEval LSAT AR | Soft Reasoning | Qwen 2 7b | 27.0 | 23.9 |
| AGIEval LSAT AR | Soft Reasoning | Qwen 2 72b | 29.1 | 28.3 |
| AGIEval LSAT AR | Soft Reasoning | GPT-4o Mini | 32.2 | 23.0 |
| AGIEval LSAT AR | Soft Reasoning | Gpt-4o | 37.8 | 30.0 |
| AGIEval LSAT AR | Soft Reasoning | Claude-3 Haiku | 24.8 | 23.5 |
| AGIEval LSAT AR | Soft Reasoning | Claude-3.5 Sonnet | 38.3 | 33.9 |
| AGIEval LSAT AR | Soft Reasoning | Gemini 1.5 Flash | 27.8 | 27.8 |
| AGIEval LSAT AR | Soft Reasoning | Gemini 1.5 Pro | 30.0 | 31.7 |
| BiGGen Bench | Soft Reasoning | Llama 2 7b | 61.6 | 56.8 |
| BiGGen Bench | Soft Reasoning | Mistral 7b | 70.1 | 68.1 |
| BiGGen Bench | Soft Reasoning | Llama 3.1 8b | 66.5 | 67.7 |
| BiGGen Bench | Soft Reasoning | Llama 3.1 70b | 78.9 | 76.9 |
| BiGGen Bench | Soft Reasoning | Gemma 2 9b | 64.7 | 64.5 |
| BiGGen Bench | Soft Reasoning | Phi-3 Small 8k | 69.7 | 63.0 |
| BiGGen Bench | Soft Reasoning | Qwen 2 7b | 46.2 | 69.9 |
| BiGGen Bench | Soft Reasoning | Qwen 2 72b | 74.3 | 79.9 |
| BiGGen Bench | Soft Reasoning | GPT-4o Mini | 70.3 | 77.7 |
| BiGGen Bench | Soft Reasoning | Gpt-4o | 86.0 | 82.0 |
| BiGGen Bench | Soft Reasoning | Claude-3 Haiku | 80.0 | 80.0 |
| BiGGen Bench | Soft Reasoning | Claude-3.5 Sonnet | 91.4 | 79.3 |
| BiGGen Bench | Soft Reasoning | Gemini 1.5 Flash | 73.9 | 68.5 |
| BiGGen Bench | Soft Reasoning | Gemini 1.5 Pro | 78.7 | 67.1 |
| Winogrande | Commonsense | Llama 2 7b | 49.9 | 50.4 |
| Winogrande | Commonsense | Mistral 7b | 60.4 | 56.5 |
| Winogrande | Commonsense | Llama 3.1 8b | 66.5 | 63.3 |
| Winogrande | Commonsense | Llama 3.1 70b | 84.2 | 81.2 |
| Winogrande | Commonsense | Gemma 2 9b | 68.7 | 67.7 |
| Winogrande | Commonsense | Phi-3 Small 8k | 81.5 | 81.6 |
| Winogrande | Commonsense | Qwen 2 7b | 67.1 | 60.7 |
| Winogrande | Commonsense | Qwen 2 72b | 81.9 | 80.7 |
| Winogrande | Commonsense | GPT-4o Mini | 79.2 | 71.9 |
| Winogrande | Commonsense | Gpt-4o | 89.7 | 86.5 |
| Winogrande | Commonsense | Claude-3 Haiku | 70.7 | 66.2 |
| Winogrande | Commonsense | Claude-3.5 Sonnet | 89.4 | 85.7 |
| Winogrande | Commonsense | Gemini 1.5 Flash | 72.5 | 74.8 |
| Winogrande | Commonsense | Gemini 1.5 Pro | 75.5 | 78.3 |
| MMLU | Knowledge | Llama 2 7b | 46.3 | 41.7 |
| MMLU | Knowledge | Mistral 7b | 60.5 | 56.5 |
| MMLU | Knowledge | Llama 3.1 8b | 72.6 | 67.5 |
| MMLU | Knowledge | Llama 3.1 70b | 85.0 | 83.2 |
| MMLU | Knowledge | Gemma 2 9b | 73.8 | 71.4 |
| MMLU | Knowledge | Phi-3 Small 8k | 76.3 | 73.6 |
| MMLU | Knowledge | Qwen 2 7b | 67.0 | 64.5 |
| MMLU | Knowledge | Qwen 2 72b | 81.3 | 77.8 |
| MMLU | Knowledge | GPT-4o Mini | 79.9 | 74.8 |
| MMLU | Knowledge | Gpt-4o | 87.5 | 83.4 |
| MMLU | Knowledge | Claude-3 Haiku | 72.2 | 68.4 |
| MMLU | Knowledge | Claude-3.5 Sonnet | 87.2 | 84.0 |
| MMLU | Knowledge | Gemini 1.5 Flash | 76.3 | 74.7 |
| MMLU | Knowledge | Gemini 1.5 Pro | 81.3 | 81.1 |
| StrategyQA | Commonsense | Llama 2 7b | 39.5 | 31.2 |
| StrategyQA | Commonsense | Mistral 7b | 66.1 | 55.8 |
| StrategyQA | Commonsense | Llama 3.1 8b | 73.7 | 68.6 |
| StrategyQA | Commonsense | Llama 3.1 70b | 85.3 | 83.8 |
| StrategyQA | Commonsense | Gemma 2 9b | 73.7 | 66.4 |
| StrategyQA | Commonsense | Phi-3 Small 8k | 72.3 | 66.0 |
| StrategyQA | Commonsense | Qwen 2 7b | 63.2 | 54.8 |
| StrategyQA | Commonsense | Qwen 2 72b | 81.7 | 76.9 |

Table 7: Zero-shot accuracy for direct answering and CoT prompts on all datasets

| Dataset | Type | Model | zero-shot CoT accuracy | zero-shot DA accuracy |
|---------|------|-------|------------------------|------------------------|
| StrategyQA | Commonsense | GPT-4o Mini | 82.2 | 84.5 |
| StrategyQA | Commonsense | Gpt-4o | 84.5 | 85.5 |
| StrategyQA | Commonsense | Claude-3 Haiku | 73.4 | 65.0 |
| StrategyQA | Commonsense | Claude-3.5 Sonnet | 80.1 | 76.3 |
| StrategyQA | Commonsense | Gemini 1.5 Flash | 72.5 | 75.2 |
| StrategyQA | Commonsense | Gemini 1.5 Pro | 74.0 | 71.4 |
| MuSR Object Placements | Soft Reasoning | Llama 2 7b | 36.3 | 30.5 |
| MuSR Object Placements | Soft Reasoning | Mistral 7b | 50.8 | 43.4 |
| MuSR Object Placements | Soft Reasoning | Llama 3.1 8b | 55.5 | 53.5 |
| MuSR Object Placements | Soft Reasoning | Llama 3.1 70b | 65.6 | 43.8 |
| MuSR Object Placements | Soft Reasoning | Gemma 2 9b | 63.3 | 57.0 |
| MuSR Object Placements | Soft Reasoning | Phi-3 Small 8k | 53.1 | 55.1 |
| MuSR Object Placements | Soft Reasoning | Qwen 2 7b | 48.8 | 48.4 |
| MuSR Object Placements | Soft Reasoning | Qwen 2 72b | 61.7 | 45.7 |
| MuSR Object Placements | Soft Reasoning | GPT-4o Mini | 59.0 | 55.0 |
| MuSR Object Placements | Soft Reasoning | Gpt-4o | 67.6 | 45.3 |
| MuSR Object Placements | Soft Reasoning | Claude-3 Haiku | 46.9 | 52.3 |
| MuSR Object Placements | Soft Reasoning | Claude-3.5 Sonnet | 69.5 | 51.2 |
| MuSR Object Placements | Soft Reasoning | Gemini 1.5 Flash | 61.7 | 56.2 |
| MuSR Object Placements | Soft Reasoning | Gemini 1.5 Pro | 66.4 | 50.0 |
| FOLIO | Symbolic | Llama 2 7b | 36.5 | 33.0 |
| FOLIO | Symbolic | Mistral 7b | 50.7 | 41.9 |
| FOLIO | Symbolic | Llama 3.1 8b | 58.6 | 56.7 |
| FOLIO | Symbolic | Llama 3.1 70b | 70.9 | 69.0 |
| FOLIO | Symbolic | Gemma 2 9b | 66.0 | 55.7 |
| FOLIO | Symbolic | Phi-3 Small 8k | 68.0 | 59.6 |
| FOLIO | Symbolic | Qwen 2 7b | 60.6 | 51.2 |
| FOLIO | Symbolic | Qwen 2 72b | 65.0 | 65.0 |
| FOLIO | Symbolic | GPT-4o Mini | 65.0 | 58.1 |
| FOLIO | Symbolic | Gpt-4o | 79.8 | 62.6 |
| FOLIO | Symbolic | Claude-3 Haiku | 61.6 | 48.8 |
| FOLIO | Symbolic | Claude-3.5 Sonnet | 73.9 | 68.5 |
| FOLIO | Symbolic | Gemini 1.5 Flash | 74.9 | 69.5 |
| FOLIO | Symbolic | Gemini 1.5 Pro | 73.9 | 74.4 |
| ContextHub Deductive L2 | Symbolic | Llama 2 7b | 34.8 | 12.6 |
| ContextHub Deductive L2 | Symbolic | Mistral 7b | 48.8 | 55.1 |
| ContextHub Deductive L2 | Symbolic | Llama 3.1 8b | 52.8 | 21.5 |
| ContextHub Deductive L2 | Symbolic | Llama 3.1 70b | 50.0 | 41.1 |
| ContextHub Deductive L2 | Symbolic | Gemma 2 9b | 50.0 | 43.0 |
| ContextHub Deductive L2 | Symbolic | Phi-3 Small 8k | 52.4 | 49.1 |
| ContextHub Deductive L2 | Symbolic | Qwen 2 7b | 51.3 | 39.8 |
| ContextHub Deductive L2 | Symbolic | Qwen 2 72b | 52.8 | 44.0 |
| ContextHub Deductive L2 | Symbolic | GPT-4o Mini | 47.0 | 42.0 |
| ContextHub Deductive L2 | Symbolic | Gpt-4o | 54.5 | 45.6 |
| ContextHub Deductive L2 | Symbolic | Claude-3 Haiku | 45.2 | 41.8 |
| ContextHub Deductive L2 | Symbolic | Claude-3.5 Sonnet | 53.0 | 46.2 |
| ContextHub Deductive L2 | Symbolic | Gemini 1.5 Flash | 45.0 | 39.5 |
| ContextHub Deductive L2 | Symbolic | Gemini 1.5 Pro | 57.3 | 43.3 |
| ContextHub Abductive L2 | Symbolic | Llama 2 7b | 34.3 | 31.9 |
| ContextHub Abductive L2 | Symbolic | Mistral 7b | 34.0 | 25.7 |
| ContextHub Abductive L2 | Symbolic | Llama 3.1 8b | 41.3 | 37.3 |
| ContextHub Abductive L2 | Symbolic | Llama 3.1 70b | 51.0 | 44.4 |
| ContextHub Abductive L2 | Symbolic | Gemma 2 9b | 41.5 | 32.9 |
| ContextHub Abductive L2 | Symbolic | Phi-3 Small 8k | 44.3 | 32.8 |
| ContextHub Abductive L2 | Symbolic | Qwen 2 7b | 37.8 | 33.4 |
| ContextHub Abductive L2 | Symbolic | Qwen 2 72b | 45.5 | 32.2 |
| ContextHub Abductive L2 | Symbolic | GPT-4o Mini | 65.0 | 55.0 |
| ContextHub Abductive L2 | Symbolic | Gpt-4o | 57.5 | 46.8 |
| ContextHub Abductive L2 | Symbolic | Claude-3 Haiku | 37.0 | 31.4 |
| ContextHub Abductive L2 | Symbolic | Claude-3.5 Sonnet | 56.8 | 40.4 |
| ContextHub Abductive L2 | Symbolic | Gemini 1.5 Flash | 53.1 | 32.2 |
| ContextHub Abductive L2 | Symbolic | Gemini 1.5 Pro | 53.5 | 43.7 |
| MMLU Pro | Knowledge | Llama 2 7b | 19.9 | 19.6 |
| MMLU Pro | Knowledge | Mistral 7b | 31.6 | 28.4 |
| MMLU Pro | Knowledge | Llama 3.1 8b | 44.8 | 38.0 |
| MMLU Pro | Knowledge | Llama 3.1 70b | 64.9 | 55.0 |
| MMLU Pro | Knowledge | Gemma 2 9b | 48.1 | 42.7 |
| MMLU Pro | Knowledge | Phi-3 Small 8k | 54.8 | 43.7 |
| MMLU Pro | Knowledge | Qwen 2 7b | 45.0 | 36.2 |
| MMLU Pro | Knowledge | Qwen 2 72b | 62.8 | 44.3 |
| MMLU Pro | Knowledge | GPT-4o Mini | 62.3 | 42.6 |
| MMLU Pro | Knowledge | Gpt-4o | 72.1 | 55.0 |
| MMLU Pro | Knowledge | Claude-3 Haiku | 47.6 | 39.0 |
| MMLU Pro | Knowledge | Claude-3.5 Sonnet | 73.4 | 57.2 |
| MMLU Pro | Knowledge | Gemini 1.5 Flash | 58.5 | 47.2 |
| MMLU Pro | Knowledge | Gemini 1.5 Pro | 65.3 | 57.4 |
| MuSR Murder Mysteries | Soft Reasoning | Llama 2 7b | 50.0 | 50.0 |

Table 7: Zero-shot accuracy for direct answering and CoT prompts on all datasets

| Dataset | Type | Model | zero-shot CoT accuracy | zero-shot DA accuracy |
|---|---|---|---|---|
| MuSR Murder Mysteries | Soft Reasoning | Mistral 7b | 62.8 | 55.6 |
| MuSR Murder Mysteries | Soft Reasoning | Llama 3.1 8b | 70.4 | 57.2 |
| MuSR Murder Mysteries | Soft Reasoning | Llama 3.1 70b | 73.6 | 69.6 |
| MuSR Murder Mysteries | Soft Reasoning | Gemma 2 9b | 76.8 | 61.6 |
| MuSR Murder Mysteries | Soft Reasoning | Phi-3 Small 8k | 61.6 | 58.8 |
| MuSR Murder Mysteries | Soft Reasoning | Qwen 2 7b | 59.2 | 53.2 |
| MuSR Murder Mysteries | Soft Reasoning | Qwen 2 72b | 80.8 | 64.4 |
| MuSR Murder Mysteries | Soft Reasoning | GPT-4o Mini | 71.2 | 63.6 |
| MuSR Murder Mysteries | Soft Reasoning | Gpt-4o | 87.6 | 70.8 |
| MuSR Murder Mysteries | Soft Reasoning | Claude-3 Haiku | 62.4 | 56.8 |
| MuSR Murder Mysteries | Soft Reasoning | Claude-3.5 Sonnet | 85.2 | 70.4 |
| MuSR Murder Mysteries | Soft Reasoning | Gemini 1.5 Flash | 70.8 | 58.4 |
| MuSR Murder Mysteries | Soft Reasoning | Gemini 1.5 Pro | 77.6 | 64.0 |
| ContextHub Deductive L1 | Symbolic | Llama 2 7b | 47.7 | 8.3 |
| ContextHub Deductive L1 | Symbolic | Mistral 7b | 50.3 | 67.3 |
| ContextHub Deductive L1 | Symbolic | Llama 3.1 8b | 50.7 | 23.3 |
| ContextHub Deductive L1 | Symbolic | Llama 3.1 70b | 53.8 | 40.7 |
| ContextHub Deductive L1 | Symbolic | Gemma 2 9b | 56.3 | 39.2 |
| ContextHub Deductive L1 | Symbolic | Phi-3 Small 8k | 54.8 | 50.2 |
| ContextHub Deductive L1 | Symbolic | Qwen 2 7b | 59.3 | 43.3 |
| ContextHub Deductive L1 | Symbolic | Qwen 2 72b | 51.5 | 44.0 |
| ContextHub Deductive L1 | Symbolic | GPT-4o Mini | 49.3 | 41.5 |
| ContextHub Deductive L1 | Symbolic | Gpt-4o | 59.3 | 49.0 |
| ContextHub Deductive L1 | Symbolic | Claude-3 Haiku | 50.5 | 39.7 |
| ContextHub Deductive L1 | Symbolic | Claude-3.5 Sonnet | 54.5 | 47.0 |
| ContextHub Deductive L1 | Symbolic | Gemini 1.5 Flash | 47.3 | 38.5 |
| ContextHub Deductive L1 | Symbolic | Gemini 1.5 Pro | 57.3 | 46.0 |
| ContextHub Abductive L1 | Symbolic | Llama 2 7b | 29.4 | 16.4 |
| ContextHub Abductive L1 | Symbolic | Mistral 7b | 46.9 | 25.8 |
| ContextHub Abductive L1 | Symbolic | Llama 3.1 8b | 43.6 | 24.2 |
| ContextHub Abductive L1 | Symbolic | Llama 3.1 70b | 55.3 | 43.9 |
| ContextHub Abductive L1 | Symbolic | Gemma 2 9b | 61.9 | 58.9 |
| ContextHub Abductive L1 | Symbolic | Phi-3 Small 8k | 62.5 | 60.3 |
| ContextHub Abductive L1 | Symbolic | Qwen 2 7b | 52.2 | 47.5 |
| ContextHub Abductive L1 | Symbolic | Qwen 2 72b | 61.9 | 45.0 |
| ContextHub Abductive L1 | Symbolic | GPT-4o Mini | 61.1 | 42.2 |
| ContextHub Abductive L1 | Symbolic | Gpt-4o | 74.2 | 65.6 |
| ContextHub Abductive L1 | Symbolic | Claude-3 Haiku | 35.3 | 22.8 |
| ContextHub Abductive L1 | Symbolic | Claude-3.5 Sonnet | 80.8 | 60.3 |
| ContextHub Abductive L1 | Symbolic | Gemini 1.5 Flash | 66.4 | 47.2 |
| ContextHub Abductive L1 | Symbolic | Gemini 1.5 Pro | 62.2 | 60.0 |
| Big-Bench Hard | Symbolic | Llama 2 7b | 29.8 | 31.9 |
| Big-Bench Hard | Symbolic | Mistral 7b | 39.3 | 35.1 |
| Big-Bench Hard | Symbolic | Llama 3.1 8b | 62.8 | 45.6 |
| Big-Bench Hard | Symbolic | Llama 3.1 70b | 78.9 | 54.8 |
| Big-Bench Hard | Symbolic | Gemma 2 9b | 58.7 | 50.8 |
| Big-Bench Hard | Symbolic | Phi-3 Small 8k | 70.0 | 55.1 |
| Big-Bench Hard | Symbolic | Qwen 2 7b | 52.6 | 47.6 |
| Big-Bench Hard | Symbolic | Qwen 2 72b | 75.1 | 59.0 |
| Big-Bench Hard | Symbolic | GPT-4o Mini | 77.7 | 49.7 |
| Big-Bench Hard | Symbolic | Gpt-4o | 84.6 | 64.5 |
| Big-Bench Hard | Symbolic | Claude-3 Haiku | 62.4 | 47.3 |
| Big-Bench Hard | Symbolic | Claude-3.5 Sonnet | 83.6 | 56.9 |
| Big-Bench Hard | Symbolic | Gemini 1.5 Flash | 71.3 | 55.4 |
| Big-Bench Hard | Symbolic | Gemini 1.5 Pro | 71.6 | 50.3 |
| MATH | Mathematical | Llama 2 7b | 4.2 | 4.0 |
| MATH | Mathematical | Mistral 7b | 12.4 | 6.1 |
| MATH | Mathematical | Llama 3.1 8b | 47.2 | 13.8 |
| MATH | Mathematical | Llama 3.1 70b | 64.4 | 22.8 |
| MATH | Mathematical | Gemma 2 9b | 45.6 | 19.1 |
| MATH | Mathematical | Phi-3 Small 8k | 43.2 | 18.5 |
| MATH | Mathematical | Qwen 2 7b | 53.7 | 13.3 |
| MATH | Mathematical | Qwen 2 72b | 63.5 | 23.8 |
| MATH | Mathematical | GPT-4o Mini | 69.6 | 24.3 |
| MATH | Mathematical | Gpt-4o | 73.3 | 35.2 |
| MATH | Mathematical | Claude-3 Haiku | 32.7 | 17.4 |
| MATH | Mathematical | Claude-3.5 Sonnet | 63.8 | 34.6 |
| MATH | Mathematical | Gemini 1.5 Flash | 54.5 | 31.3 |
| MATH | Mathematical | Gemini 1.5 Pro | 62.1 | 39.4 |
| GSM8k-Hard | Mathematical | Llama 2 7b | 6.7 | 1.8 |
| GSM8k-Hard | Mathematical | Mistral 7b | 21.0 | 3.0 |
| GSM8k-Hard | Mathematical | Llama 3.1 8b | 34.4 | 6.0 |
| GSM8k-Hard | Mathematical | Llama 3.1 70b | 46.6 | 14.0 |
| GSM8k-Hard | Mathematical | Gemma 2 9b | 40.9 | 8.8 |
| GSM8k-Hard | Mathematical | Phi-3 Small 8k | 33.0 | 6.9 |
| GSM8k-Hard | Mathematical | Qwen 2 7b | 48.4 | 5.0 |
| GSM8k-Hard | Mathematical | Qwen 2 72b | 54.8 | 13.7 |

Table 7: Zero-shot accuracy for direct answering and CoT prompts on all datasets

| Dataset | Type | Model | zero-shot CoT accuracy | zero-shot DA accuracy |
|---|---|---|---|---|
| GSM8k-Hard | Mathematical | GPT-4o Mini | 53.9 | 11.7 |
| GSM8k-Hard | Mathematical | Gpt-4o | 60.3 | 26.0 |
| GSM8k-Hard | Mathematical | Claude-3 Haiku | 45.3 | 9.6 |
| GSM8k-Hard | Mathematical | Claude-3.5 Sonnet | 50.8 | 32.3 |
| GSM8k-Hard | Mathematical | Gemini 1.5 Flash | 54.6 | 16.2 |
| GSM8k-Hard | Mathematical | Gemini 1.5 Pro | 58.2 | 26.2 |
| GSM8k | Mathematical | Llama 2 7b | 29.6 | 6.9 |
| GSM8k | Mathematical | Mistral 7b | 59.2 | 10.2 |
| GSM8k | Mathematical | Llama 3.1 8b | 85.4 | 18.5 |
| GSM8k | Mathematical | Llama 3.1 70b | 85.6 | 37.0 |
| GSM8k | Mathematical | Gemma 2 9b | 89.2 | 24.9 |
| GSM8k | Mathematical | Phi-3 Small 8k | 90.0 | 24.9 |
| GSM8k | Mathematical | Qwen 2 7b | 87.9 | 20.7 |
| GSM8k | Mathematical | Qwen 2 72b | 94.6 | 40.1 |
| GSM8k | Mathematical | GPT-4o Mini | 94.1 | 31.8 |
| GSM8k | Mathematical | Gpt-4o | 95.8 | 58.8 |
| GSM8k | Mathematical | Claude-3 Haiku | 89.4 | 22.9 |
| GSM8k | Mathematical | Claude-3.5 Sonnet | 96.1 | 62.2 |
| GSM8k | Mathematical | Gemini 1.5 Flash | 91.4 | 38.6 |
| GSM8k | Mathematical | Gemini 1.5 Pro | 92.7 | 52.4 |

## F.2 FULL FEW-SHOT RESULTS

Table 8: Few-shot accuracy for direct answering and CoT prompts on all datasets

| Dataset | Type | Model | few-shot CoT accuracy | few-shot DA accuracy |
|---|---|---|---|---|
| AGIEval LSAT RC | Soft Reasoning | Llama 2 7b | 33.1 | 38.7 |
| AGIEval LSAT RC | Soft Reasoning | Mistral 7b | 52.4 | 57.2 |
| AGIEval LSAT RC | Soft Reasoning | Llama 3.1 8b | 60.2 | 70.3 |
| AGIEval LSAT RC | Soft Reasoning | Llama 3.1 70b | 84.4 | 88.8 |
| AGIEval LSAT RC | Soft Reasoning | Gemma 2 9b | 74.3 | 79.2 |
| AGIEval LSAT RC | Soft Reasoning | Phi-3 Small 8k | 63.2 | 65.1 |
| AGIEval LSAT RC | Soft Reasoning | Qwen 2 7b | 61.7 | 68.8 |
| AGIEval LSAT RC | Soft Reasoning | Qwen 2 72b | 85.9 | 85.9 |
| AGIEval LSAT RC | Soft Reasoning | GPT-4o Mini | 77.3 | 71.4 |
| AGIEval LSAT RC | Soft Reasoning | Gemini 1.5 Flash | 79.2 | 81.8 |
| AGIEval LSAT LR | Soft Reasoning | Llama 2 7b | 33.7 | 34.7 |
| AGIEval LSAT LR | Soft Reasoning | Mistral 7b | 46.1 | 48.0 |
| AGIEval LSAT LR | Soft Reasoning | Llama 3.1 8b | 55.7 | 58.0 |
| AGIEval LSAT LR | Soft Reasoning | Llama 3.1 70b | 83.3 | 85.1 |
| AGIEval LSAT LR | Soft Reasoning | Gemma 2 9b | 65.7 | 68.2 |
| AGIEval LSAT LR | Soft Reasoning | Phi-3 Small 8k | 64.7 | 59.2 |
| AGIEval LSAT LR | Soft Reasoning | Qwen 2 7b | 54.1 | 61.2 |
| AGIEval LSAT LR | Soft Reasoning | Qwen 2 72b | 77.5 | 79.6 |
| AGIEval LSAT LR | Soft Reasoning | GPT-4o Mini | 68.4 | 64.5 |
| AGIEval LSAT LR | Soft Reasoning | Gemini 1.5 Flash | 68.6 | 72.9 |
| GPQA | Mathematical | Mistral 7b | 23.0 | 25.9 |
| GPQA | Mathematical | Llama 3.1 8b | 22.1 | 27.2 |
| GPQA | Mathematical | Llama 3.1 70b | 24.8 | 24.3 |
| GPQA | Mathematical | Gemma 2 9b | 19.9 | 22.3 |
| GPQA | Mathematical | Phi-3 Small 8k | 23.9 | 22.5 |
| GPQA | Mathematical | Qwen 2 7b | 23.4 | 21.2 |
| GPQA | Mathematical | Qwen 2 72b | 22.8 | 19.9 |
| GPQA | Mathematical | GPT-4o Mini | 20.0 | 20.0 |
| GPQA | Mathematical | Gemini 1.5 Flash | 21.9 | 24.6 |
| CommonsenseQA | Commonsense | Llama 2 7b | 18.2 | 19.2 |
| CommonsenseQA | Commonsense | Mistral 7b | 73.6 | 70.4 |
| CommonsenseQA | Commonsense | Llama 3.1 8b | 74.0 | 76.5 |
| CommonsenseQA | Commonsense | Llama 3.1 70b | 84.7 | 84.6 |
| CommonsenseQA | Commonsense | Gemma 2 9b | 81.8 | 80.8 |
| CommonsenseQA | Commonsense | Phi-3 Small 8k | 80.8 | 80.4 |
| CommonsenseQA | Commonsense | Qwen 2 7b | 80.3 | 72.9 |
| CommonsenseQA | Commonsense | Qwen 2 72b | 88.4 | 87.8 |
| CommonsenseQA | Commonsense | GPT-4o Mini | 84.7 | 84.7 |
| CommonsenseQA | Commonsense | Gemini 1.5 Flash | 81.7 | 83.3 |
| AGIEval LSAT AR | Soft Reasoning | Llama 2 7b | 19.6 | 18.7 |
| AGIEval LSAT AR | Soft Reasoning | Mistral 7b | 20.9 | 22.6 |
| AGIEval LSAT AR | Soft Reasoning | Llama 3.1 8b | 24.8 | 26.1 |
| AGIEval LSAT AR | Soft Reasoning | Llama 3.1 70b | 36.1 | 30.9 |
| AGIEval LSAT AR | Soft Reasoning | Gemma 2 9b | 22.2 | 28.7 |
| AGIEval LSAT AR | Soft Reasoning | Phi-3 Small 8k | 27.8 | 20.0 |
| AGIEval LSAT AR | Soft Reasoning | Qwen 2 7b | 24.3 | 23.0 |
| AGIEval LSAT AR | Soft Reasoning | Qwen 2 72b | 27.0 | 30.0 |

Table 8: Few-shot accuracy for direct answering and CoT prompts on all datasets

| Dataset | Type | Model | few-shot CoT accuracy | few-shot DA accuracy |
|---|---|---|---|---|
| AGIEval LSAT AR | Soft Reasoning | GPT-4o Mini | 28.7 | 26.1 |
| AGIEval LSAT AR | Soft Reasoning | Gemini 1.5 Flash | 28.3 | 20.4 |
| MMLU | Knowledge | Llama 2 7b | 49.0 | 42.8 |
| MMLU | Knowledge | Mistral 7b | 63.0 | 57.0 |
| MMLU | Knowledge | Llama 3.1 8b | 71.7 | 69.3 |
| MMLU | Knowledge | Llama 3.1 70b | 84.3 | 83.7 |
| MMLU | Knowledge | Gemma 2 9b | 74.7 | 72.4 |
| MMLU | Knowledge | Phi-3 Small 8k | 77.3 | 75.2 |
| MMLU | Knowledge | Qwen 2 7b | 69.9 | 68.6 |
| MMLU | Knowledge | Qwen 2 72b | 82.7 | 81.8 |
| MMLU | Knowledge | GPT-4o Mini | 82.3 | 77.8 |
| MMLU | Knowledge | Gemini 1.5 Flash | 78.1 | 79.0 |
| StrategyQA | Commonsense | Llama 2 7b | 57.9 | 30.9 |
| StrategyQA | Commonsense | Mistral 7b | 70.7 | 72.0 |
| StrategyQA | Commonsense | Llama 3.1 8b | 74.4 | 65.8 |
| StrategyQA | Commonsense | Llama 3.1 70b | 87.1 | 84.2 |
| StrategyQA | Commonsense | Gemma 2 9b | 77.1 | 73.3 |
| StrategyQA | Commonsense | Phi-3 Small 8k | 75.0 | 71.1 |
| StrategyQA | Commonsense | Qwen 2 7b | 71.9 | 58.9 |
| StrategyQA | Commonsense | Qwen 2 72b | 83.2 | 80.1 |
| StrategyQA | Commonsense | GPT-4o Mini | 83.0 | 86.2 |
| StrategyQA | Commonsense | Gemini 1.5 Flash | 77.0 | 80.3 |
| ContextHub Abductive L2 | Symbolic | Llama 2 7b | 36.2 | 35.0 |
| ContextHub Abductive L2 | Symbolic | Mistral 7b | 33.8 | 30.0 |
| ContextHub Abductive L2 | Symbolic | Llama 3.1 8b | 32.7 | 36.1 |
| ContextHub Abductive L2 | Symbolic | Llama 3.1 70b | 54.6 | 51.2 |
| ContextHub Abductive L2 | Symbolic | Gemma 2 9b | 44.8 | 33.2 |
| ContextHub Abductive L2 | Symbolic | Phi-3 Small 8k | 49.8 | 34.2 |
| ContextHub Abductive L2 | Symbolic | Qwen 2 7b | 39.6 | 35.0 |
| ContextHub Abductive L2 | Symbolic | Qwen 2 72b | 54.7 | 34.9 |
| ContextHub Abductive L2 | Symbolic | GPT-4o Mini | 62.0 | 60.0 |
| ContextHub Abductive L2 | Symbolic | Gemini 1.5 Flash | 48.6 | 47.8 |
| ContextHub Abductive L1 | Symbolic | Llama 2 7b | 21.4 | 16.7 |
| ContextHub Abductive L1 | Symbolic | Mistral 7b | 23.6 | 21.7 |
| ContextHub Abductive L1 | Symbolic | Llama 3.1 8b | 40.0 | 36.1 |
| ContextHub Abductive L1 | Symbolic | Llama 3.1 70b | 62.2 | 58.9 |
| ContextHub Abductive L1 | Symbolic | Gemma 2 9b | 48.9 | 59.4 |
| ContextHub Abductive L1 | Symbolic | Phi-3 Small 8k | 59.2 | 56.4 |
| ContextHub Abductive L1 | Symbolic | Qwen 2 7b | 48.6 | 38.9 |
| ContextHub Abductive L1 | Symbolic | Qwen 2 72b | 53.3 | 56.1 |
| ContextHub Abductive L1 | Symbolic | GPT-4o Mini | 77.2 | 59.2 |
| ContextHub Abductive L1 | Symbolic | Gemini 1.5 Flash | 79.7 | 68.6 |
| MuSR Murder Mysteries | Soft Reasoning | Mistral 7b | 62.0 | 56.4 |
| MuSR Murder Mysteries | Soft Reasoning | Llama 3.1 8b | 61.6 | 61.2 |
| MuSR Murder Mysteries | Soft Reasoning | Llama 3.1 70b | 73.2 | 68.0 |
| MuSR Murder Mysteries | Soft Reasoning | Gemma 2 9b | 81.6 | 62.0 |
| MuSR Murder Mysteries | Soft Reasoning | Phi-3 Small 8k | 62.0 | 53.6 |
| MuSR Murder Mysteries | Soft Reasoning | Qwen 2 7b | 56.0 | 55.6 |
| MuSR Murder Mysteries | Soft Reasoning | Qwen 2 72b | 80.4 | 66.0 |
| MuSR Murder Mysteries | Soft Reasoning | GPT-4o Mini | 76.0 | 69.6 |
| MuSR Murder Mysteries | Soft Reasoning | Gemini 1.5 Flash | 70.0 | 66.4 |
| MuSR Team Allocations | Soft Reasoning | Mistral 7b | 42.8 | 43.2 |
| MuSR Team Allocations | Soft Reasoning | Llama 3.1 8b | 59.6 | 51.6 |
| MuSR Team Allocations | Soft Reasoning | Llama 3.1 70b | 89.2 | 63.6 |
| MuSR Team Allocations | Soft Reasoning | Gemma 2 9b | 48.4 | 45.6 |
| MuSR Team Allocations | Soft Reasoning | Phi-3 Small 8k | 66.0 | 46.4 |
| MuSR Team Allocations | Soft Reasoning | Qwen 2 7b | 34.0 | 40.8 |
| MuSR Team Allocations | Soft Reasoning | Qwen 2 72b | 56.0 | 66.4 |
| MuSR Team Allocations | Soft Reasoning | GPT-4o Mini | 75.6 | 60.0 |
| MuSR Team Allocations | Soft Reasoning | Gemini 1.5 Flash | 90.0 | 54.4 |
| MMLU Pro | Knowledge | Llama 2 7b | 21.5 | 20.4 |
| MMLU Pro | Knowledge | Mistral 7b | 34.8 | 26.7 |
| MMLU Pro | Knowledge | Llama 3.1 8b | 44.7 | 38.0 |
| MMLU Pro | Knowledge | Llama 3.1 70b | 64.4 | 55.1 |
| MMLU Pro | Knowledge | Gemma 2 9b | 48.5 | 42.4 |
| MMLU Pro | Knowledge | Phi-3 Small 8k | 54.8 | 43.2 |
| MMLU Pro | Knowledge | Qwen 2 7b | 46.6 | 39.0 |
| MMLU Pro | Knowledge | Qwen 2 72b | 62.5 | 51.6 |
| MMLU Pro | Knowledge | GPT-4o Mini | 63.0 | 45.0 |
| MMLU Pro | Knowledge | Gemini 1.5 Flash | 59.4 | 50.6 |
| MuSR Object Placements | Soft Reasoning | Mistral 7b | 55.5 | 41.0 |
| MuSR Object Placements | Soft Reasoning | Llama 3.1 8b | 66.8 | 50.4 |
| MuSR Object Placements | Soft Reasoning | Llama 3.1 70b | 67.2 | 57.4 |
| MuSR Object Placements | Soft Reasoning | Gemma 2 9b | 68.0 | 58.2 |
| MuSR Object Placements | Soft Reasoning | Phi-3 Small 8k | 62.1 | 51.6 |
| MuSR Object Placements | Soft Reasoning | Qwen 2 7b | 46.9 | 43.8 |
| MuSR Object Placements | Soft Reasoning | Qwen 2 72b | 66.4 | 43.0 |

Table 8: Few-shot accuracy for direct answering and CoT prompts on all datasets

| Dataset | Type | Model | few-shot CoT accuracy | few-shot DA accuracy |
|---|---|---|---|---|
| MuSR Object Placements | Soft Reasoning | GPT-4o Mini | 67.0 | 47.0 |
| MuSR Object Placements | Soft Reasoning | Gemini 1.5 Flash | 73.0 | 54.7 |
| ContextHub Deductive L2 | Symbolic | Llama 2 7b | 34.7 | 15.0 |
| ContextHub Deductive L2 | Symbolic | Mistral 7b | 63.8 | 51.4 |
| ContextHub Deductive L2 | Symbolic | Llama 3.1 8b | 76.1 | 27.3 |
| ContextHub Deductive L2 | Symbolic | Llama 3.1 70b | 82.6 | 53.6 |
| ContextHub Deductive L2 | Symbolic | Gemma 2 9b | 61.9 | 47.6 |
| ContextHub Deductive L2 | Symbolic | Phi-3 Small 8k | 61.5 | 54.0 |
| ContextHub Deductive L2 | Symbolic | Qwen 2 7b | 55.3 | 36.4 |
| ContextHub Deductive L2 | Symbolic | Qwen 2 72b | 80.2 | 54.0 |
| ContextHub Deductive L2 | Symbolic | GPT-4o Mini | 59.0 | 41.0 |
| ContextHub Deductive L2 | Symbolic | Gemini 1.5 Flash | 90.2 | 42.5 |
| ContextHub Deductive L1 | Symbolic | Llama 2 7b | 34.7 | 16.0 |
| ContextHub Deductive L1 | Symbolic | Mistral 7b | 46.2 | 59.2 |
| ContextHub Deductive L1 | Symbolic | Llama 3.1 8b | 73.0 | 23.0 |
| ContextHub Deductive L1 | Symbolic | Llama 3.1 70b | 67.5 | 50.0 |
| ContextHub Deductive L1 | Symbolic | Gemma 2 9b | 66.0 | 45.7 |
| ContextHub Deductive L1 | Symbolic | Phi-3 Small 8k | 74.8 | 51.8 |
| ContextHub Deductive L1 | Symbolic | Qwen 2 7b | 58.8 | 37.5 |
| ContextHub Deductive L1 | Symbolic | Qwen 2 72b | 70.7 | 42.8 |
| ContextHub Deductive L1 | Symbolic | GPT-4o Mini | 59.2 | 44.3 |
| ContextHub Deductive L1 | Symbolic | Gemini 1.5 Flash | 89.3 | 49.8 |
| MATH | Mathematical | Llama 2 7b | 4.7 | 3.9 |
| MATH | Mathematical | Mistral 7b | 13.7 | 7.1 |
| MATH | Mathematical | Llama 3.1 8b | 41.2 | 14.2 |
| MATH | Mathematical | Llama 3.1 70b | 61.9 | 24.2 |
| MATH | Mathematical | Gemma 2 9b | 47.5 | 19.8 |
| MATH | Mathematical | Phi-3 Small 8k | 42.4 | 18.9 |
| MATH | Mathematical | Qwen 2 7b | 55.0 | 15.0 |
| MATH | Mathematical | Qwen 2 72b | 65.3 | 26.2 |
| MATH | Mathematical | GPT-4o Mini | 71.7 | 24.6 |
| MATH | Mathematical | Gemini 1.5 Flash | 54.7 | 32.3 |
| GSM8K | Mathematical | Llama 2 7b | 29.0 | 7.7 |
| GSM8K | Mathematical | Mistral 7b | 56.2 | 12.5 |
| GSM8K | Mathematical | Llama 3.1 8b | 86.4 | 20.1 |
| GSM8K | Mathematical | Llama 3.1 70b | 96.1 | 39.1 |
| GSM8K | Mathematical | Gemma 2 9b | 89.2 | 24.9 |
| GSM8K | Mathematical | Phi-3 Small 8k | 90.4 | 24.5 |
| GSM8K | Mathematical | Qwen 2 7b | 87.6 | 21.4 |
| GSM8K | Mathematical | Qwen 2 72b | 93.2 | 40.6 |
| GSM8K | Mathematical | GPT-4o Mini | 94.2 | 32.8 |
| GSM8K | Mathematical | Gemini 1.5 Flash | 90.6 | 40.4 |

### F.3 Answer Extractor and Average Answer Span Results

In this section, we report the number of generations from each model on each dataset that our answer parser could not extract. "-1" denotes that a model was not run on a certain dataset due to context length limitations in the few-shot setting. We see that these unparseable rates are generally low across the board. The weakest models struggle on some of the most challenging datasets, but unparseable rates are all at or below 15%.

We also report the average character index of the beginning of the answer span that the answer parser extracted. Of particular note is that the direct answer prompts all return an answer within the first 60 characters, indicating that the answers are returned almost immediately, as desired. CoT completions are much longer.

## G Zoom-in: MMLU and MMLU Pro

MMLU and MMLU Pro show gains from adding CoT, but because these datasets are so broad, they defy simple characterization. We explore the performance of CoT on each category of MMLU to understand divergences in CoT performance between these domains. We list the top three categories where CoT gives the largest error reduction for Llama 3.1 8B and 70B on MMLU and MMLU Pro in Table 17. Some of these categories are explicitly mathematical in nature, as we might expect from Figure 3. We can also see that CoT is helping on categories like "business"; upon closer inspection, we found that these categories frequently involve math as well (e.g., business questions may involve computations surrounding wealth). We need to more carefully characterize MMLU at the *instance*

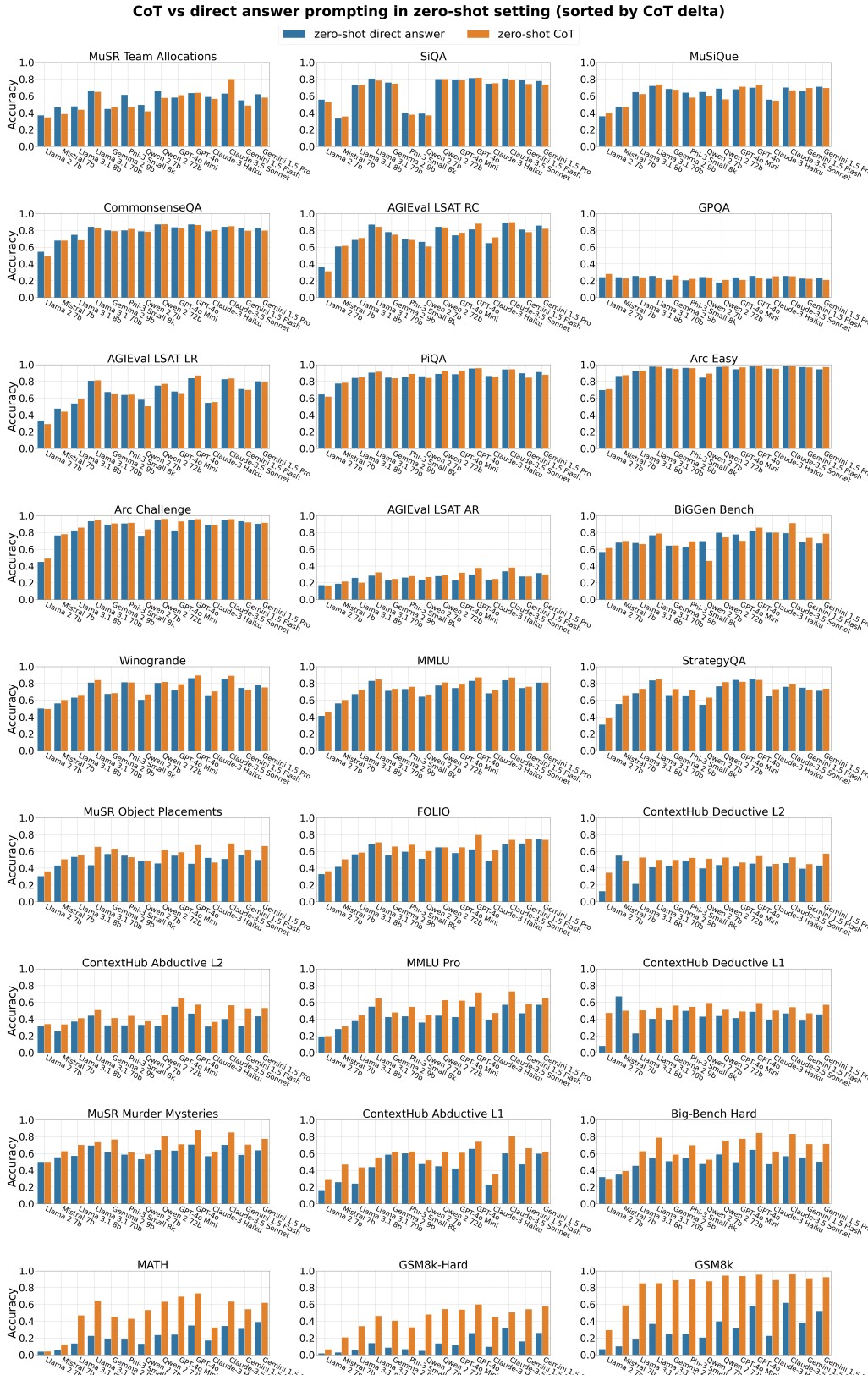

Figure 9: Performance of zero-shot direct (blue) and zero-shot CoT (orange) across datasets and models. Graphs are sorted in ascending order by median delta (CoT, direct). The datasets benefiting substantially are all symbolic or semi-symbolic in nature.

Table 9: Percentage of responses per dataset per model that our answer parser could not extract an answer for in the zero-shot direct answer setting. Prompt modifications were made to decrease these numbers. No model is above 15%.

| dataset | Meta-Llama 2 7b | Mistral 7b | Meta-Llama 3.1 8b | Meta-Llama 3.1 70b | Gemma 2 9b | Phi-3 Small 8k | Qwen 2 7b | Qwen 2 72b | GPT-4o Mini | Gpt-4o | Claude-3 Haiku | Claude-3.5 Sonnet | Gemini 1.5 Flash | Gemini 1.5 Pro |
|---|---|---|---|---|---|---|---|---|---|---|---|---|---|---|
| CommonsenseQA | 1.9 | 2.5 | 1.1 | 0.0 | 0.8 | 0.1 | 1.6 | 0.7 | 0.0 | 0.0 | 0.1 | 0.0 | 0.1 | 0.2 |
| StrategyQA | 0.0 | 1.9 | 0.1 | 0.0 | 11.7 | 0.5 | 4.9 | 2.7 | 0.0 | 0.0 | 0.0 | 0.0 | 0.0 | 0.2 |
| SiQA | 0.2 | 6.6 | 0.0 | 0.1 | 3.9 | 0.3 | 0.1 | 3.0 | 0.1 | 0.1 | 0.0 | 0.0 | 0.0 | 0.4 |
| PiQA | 0.4 | 6.0 | 0.0 | 0.1 | 3.3 | 2.1 | 0.0 | 5.5 | 0.2 | 0.0 | 0.1 | 0.0 | 0.1 | 0.9 |
| Winogrande | 0.0 | 3.0 | 0.1 | 0.0 | 2.1 | 0.2 | 5.1 | 0.4 | 0.0 | 0.0 | 0.0 | 0.0 | 0.0 | 3.6 |
| Arc Easy | 0.0 | 1.8 | 0.5 | 0.0 | 0.0 | 0.2 | 9.1 | 0.7 | 3.5 | 0.4 | 0.2 | 0.0 | 0.0 | 3.2 |
| Arc Challenge | 0.0 | 2.3 | 1.0 | 0.0 | 0.3 | 0.7 | 10.7 | 0.7 | 10.0 | 0.7 | 0.0 | 0.0 | 0.0 | 5.0 |
| AGIEval LSAT LR | 0.4 | 0.0 | 0.0 | 0.0 | 0.0 | 0.2 | 0.0 | 0.0 | 0.0 | 0.0 | 2.5 | 0.0 | 0.2 | 0.2 |
| AGIEval LSAT AR | 0.4 | 0.0 | 0.0 | 0.0 | 4.3 | 3.9 | 0.0 | 0.0 | 0.0 | 8.7 | 0.0 | 0.0 | 0.0 | 0.0 |
| AGIEval LSAT RC | 0.4 | 0.4 | 0.0 | 0.0 | 0.0 | 0.0 | 0.0 | 0.0 | 0.0 | 9.7 | 0.0 | 0.0 | 0.4 | 0.4 |
| ContextHub Deductive L1 | 0.0 | 0.0 | 0.0 | 0.0 | 1.2 | 0.0 | 2.3 | 0.0 | 0.0 | 0.0 | 0.2 | 0.0 | 0.0 | 0.2 |
| ContextHub Deductive L2 | 0.0 | 0.0 | 0.0 | 0.0 | 0.0 | 0.0 | 2.2 | 1.0 | 0.0 | 0.0 | 2.8 | 0.0 | 0.0 | 0.0 |
| ContextHub Abductive L1 | 0.0 | 0.0 | 0.0 | 0.0 | 0.0 | 0.3 | 0.0 | 0.0 | 0.0 | 0.0 | 0.0 | 0.0 | 0.0 | 0.0 |
| ContextHub Abductive L2 | 0.0 | 0.0 | 0.0 | 0.0 | 0.0 | 0.1 | 1.5 | 0.2 | 0.0 | 0.0 | 0.8 | 0.0 | 0.0 | 0.0 |
| MuSR Murder Mysteries | 0.0 | 0.0 | 0.0 | 0.0 | 0.0 | 0.0 | 0.0 | 0.0 | 0.0 | 0.0 | 0.0 | 0.0 | 0.0 | 0.0 |
| MuSR Team Allocations | 0.0 | 0.0 | 0.0 | 0.0 | 3.6 | 0.0 | 0.0 | 0.0 | 0.0 | 0.0 | 0.0 | 0.0 | 8.4 | 0.4 |
| MuSR Object Placements | 0.0 | 0.0 | 0.0 | 0.0 | 0.0 | 0.0 | 0.0 | 0.0 | 0.0 | 0.0 | 0.0 | 0.0 | 0.0 | 0.0 |
| MMLU | 0.1 | 0.0 | 0.0 | 0.0 | 0.1 | 0.2 | 3.6 | 1.2 | 0.6 | 0.0 | 1.3 | 0.3 | 0.2 | 0.7 |
| MMLU Pro | 0.7 | 1.3 | 1.0 | 0.3 | 1.0 | 3.7 | 6.8 | 12.2 | 0.4 | 0.3 | 0.4 | 0.6 | 0.6 | 0.8 |
| GPQA | 1.3 | 7.1 | 0.0 | 0.0 | 8.7 | 12.7 | 5.4 | 15.2 | 0.0 | 0.0 | 1.6 | 0.0 | 0.0 | 0.7 |
| MATH | 0.6 | 6.9 | 0.2 | 0.2 | 0.1 | 0.1 | 3.5 | 3.0 | 0.2 | 0.0 | 0.0 | 0.0 | 0.4 | 0.0 |
| GSM8k | 0.2 | 4.1 | 2.5 | 0.0 | 2.7 | 0.0 | 1.7 | 0.2 | 0.0 | 0.0 | 12.7 | 5.5 | 0.0 | 0.0 |
| BigGen Bench | 4.6 | 0.3 | 0.9 | 0.1 | 0.5 | 1.0 | 1.3 | 1.0 | 1.3 | 0.0 | 0.0 | 0.1 | 0.4 | 0.3 |
| GSM8k-Hard | 4.8 | 7.6 | 2.0 | 2.4 | 0.4 | 0.2 | 3.2 | 1.1 | 0.1 | 0.5 | 5.2 | 0.5 | 0.2 | 0.0 |
| MuSiQue | 0.1 | 0.0 | 0.0 | 0.0 | 0.0 | 0.1 | 0.0 | 0.0 | 0.0 | 0.0 | 0.0 | 0.0 | 0.2 | 0.1 |
| Folio | 4.4 | 0.0 | 0.0 | 0.0 | 0.0 | 0.0 | 3.9 | 0.0 | 0.0 | 12.3 | 0.0 | 0.0 | 0.0 | 0.5 |
| BigBench-Hard | 0.0 | 0.0 | 0.0 | 7.4 | 0.0 | 0.2 | 0.0 | 0.0 | 0.0 | 0.3 | 0.0 | 4.5 | 0.2 | 12.8 |

Table 10: Percentage of responses per dataset per model that our answer parser could not extract an answer for in the zero-shot CoT setting. Prompt modifications were made to decrease these numbers. No model is above 15%.

| dataset | Meta-Llama 2 7b | Mistral 7b | Meta-Llama 3.1 8b | Meta-Llama 3.1 70b | Gemma 2 9b | Phi-3 Small 8k | Qwen 2 7b | Qwen 2 72b | GPT-4o Mini | Gpt-4o | Claude-3 Haiku | Claude-3.5 Sonnet | Gemini 1.5 Flash | Gemini 1.5 Pro |
|---|---|---|---|---|---|---|---|---|---|---|---|---|---|---|
| CommonsenseQA | 2.9 | 1.3 | 8.6 | 0.0 | 0.6 | 0.1 | 0.0 | 0.0 | 1.6 | 0.0 | 0.2 | 0.3 | 2.4 | 2.6 |
| StrategyQA | 1.0 | 0.1 | 1.1 | 0.8 | 0.3 | 0.4 | 0.3 | 0.0 | 0.0 | 0.0 | 0.0 | 0.0 | 2.1 | 4.4 |
| SiQA | 0.8 | 1.8 | 0.3 | 0.1 | 1.6 | 0.0 | 0.1 | 0.3 | 0.0 | 0.0 | 1.4 | 0.3 | 4.6 | 4.0 |
| PiQA | 1.6 | 1.6 | 0.2 | 0.1 | 2.8 | 0.3 | 0.5 | 0.3 | 0.0 | 0.0 | 1.4 | 0.3 | 4.6 | 4.6 |
| Winogrande | 0.9 | 1.4 | 0.2 | 0.2 | 0.9 | 0.4 | 0.3 | 0.0 | 0.0 | 0.0 | 0.0 | 0.0 | 2.0 | 3.4 |
| Arc Easy | 0.2 | 0.4 | 0.2 | 0.0 | 0.5 | 1.6 | 1.6 | 0.0 | 0.5 | 0.0 | 0.0 | 0.0 | 0.4 | 0.5 |
| Arc Challenge | 0.0 | 0.7 | 0.0 | 0.0 | 0.0 | 0.0 | 0.7 | 0.0 | 0.0 | 0.0 | 0.0 | 0.0 | 0.7 | 0.7 |
| AGIEval LSAT LR | 3.3 | 2.2 | 0.0 | 0.0 | 1.2 | 0.0 | 2.0 | 0.0 | 0.0 | 0.0 | 0.0 | 0.0 | 0.8 | 0.2 |
| AGIEval LSAT AR | 4.8 | 7.0 | 6.1 | 2.2 | 5.7 | 5.2 | 4.3 | 0.4 | 1.3 | 1.3 | 0.0 | 0.4 | 4.8 | 1.7 |
| AGIEval LSAT RC | 7.1 | 1.1 | 0.0 | 0.0 | 0.7 | 3.0 | 6.7 | 0.0 | 0.0 | 0.0 | 0.0 | 0.0 | 0.0 | 0.4 |
| ContextHub Deductive L1 | 0.7 | 1.3 | 0.0 | 0.0 | 0.0 | 0.0 | 0.0 | 0.0 | 0.0 | 0.0 | 0.0 | 0.0 | 0.0 | 0.3 |
| ContextHub Deductive L2 | 0.2 | 0.4 | 0.0 | 0.0 | 0.0 | 0.0 | 0.0 | 0.0 | 0.0 | 0.0 | 0.0 | 0.0 | 0.5 | 0.4 |
| ContextHub Abductive L1 | 0.6 | 0.0 | 0.0 | 0.0 | 0.0 | 0.0 | 0.0 | 0.0 | 0.0 | 0.0 | 0.0 | 0.0 | 0.0 | 0.0 |
| ContextHub Abductive L2 | 0.0 | 0.2 | 0.1 | 0.0 | 0.0 | 0.3 | 0.0 | 0.0 | 0.0 | 0.0 | 0.0 | 0.0 | 0.5 | 0.4 |
| MuSR Murder Mysteries | 0.0 | 0.4 | 0.0 | 0.0 | 0.0 | 11.6 | 0.4 | 0.0 | 0.0 | 0.0 | 0.0 | 0.0 | 6.8 | 3.6 |
| MuSR Team Allocations | 5.2 | 3.2 | 0.8 | 0.0 | 0.8 | 0.4 | 0.4 | 0.0 | 0.0 | 0.0 | 0.0 | 0.0 | 0.0 | 0.0 |
| MuSR Object Placements | 0.0 | 1.6 | 0.0 | 0.0 | 0.4 | 0.8 | 0.0 | 0.0 | 0.0 | 0.0 | 0.0 | 0.0 | 2.0 | 0.4 |
| MMLU | 1.9 | 0.6 | 1.0 | 0.2 | 1.5 | 1.0 | 0.4 | 0.2 | 0.0 | 0.1 | 0.0 | 0.1 | 3.1 | 3.2 |
| MMLU Pro | 4.4 | 5.4 | 13.1 | 3.3 | 12.5 | 3.6 | 5.4 | 2.0 | 2.4 | 1.9 | 0.4 | 0.4 | 5.0 | 4.4 |
| GPQA | 4.5 | 10.3 | 9.4 | 1.6 | 8.5 | 1.8 | 3.8 | 0.7 | 0.0 | 0.0 | 0.0 | 0.0 | 11.8 | 15.0 |
| MATH | 1.6 | 5.5 | 8.2 | 2.5 | 2.3 | 1.6 | 3.0 | 0.4 | 0.4 | 0.5 | 0.9 | 0.0 | 1.7 | 1.0 |
| GSM8k | 1.7 | 1.4 | 0.7 | 10.5 | 0.4 | 0.6 | 0.4 | 0.0 | 0.0 | 0.0 | 0.3 | 0.0 | 0.1 | 0.1 |
| BigGen Bench | 5.0 | 0.0 | 0.5 | 0.1 | 0.5 | 0.4 | 0.3 | 9.5 | 0.0 | 0.0 | 0.0 | 0.1 | 0.4 | 0.1 |
| GSM8k-Hard | 2.1 | 8.7 | 10.2 | 4.5 | 10.7 | 3.2 | 3.5 | 1.0 | 0.8 | 0.5 | 3.0 | 1.8 | 0.4 | 2.7 |
| MuSiQue | 1.4 | 0.0 | 8.3 | 0.1 | 0.0 | 0.0 | 0.0 | 0.0 | 0.0 | 0.0 | 0.0 | 0.0 | 0.7 | 3.1 |
| Folio | 0.0 | 0.0 | 1.5 | 0.0 | 0.0 | 0.0 | 0.0 | 0.0 | 0.0 | 0.0 | 0.0 | 0.0 | 2.0 | 1.5 |
| BigBench-Hard | 3.8 | 5.4 | 1.8 | 0.4 | 1.3 | 0.1 | 0.4 | 0.3 | 0.0 | 0.0 | 0.0 | 0.0 | 1.2 | 0.9 |

Table 11: Percentage of responses per dataset per model that our answer parser could not extract an answer for in the few-shot direct answer setting. Prompt modifications were made to decrease these numbers. No model is above 15%.

| Few-shot Direct Answer Unparseable Answer Rate by Percentage | | | | | | | | | |
|---|---|---|---|---|---|---|---|---|---|
| dataset | Meta-Llama 2 7b | Mistral 7b | Meta-Llama 3.1 8b | Meta-Llama 3.1 70b | Gemma 2 9b | Phi-3 Small 8k | Qwen 2 7b | Qwen 2 72b | GPT-4o Mini | Gemini 1.5 Flash |
| CommonsenseQA | 0.0 | 0.1 | 0.2 | 0.0 | 1.3 | 0.9 | 9.9 | 1.3 | 0.0 | 0.6 |
| AGIEval LSAT LR | 6.7 | 0.0 | 0.0 | 0.0 | 0.0 | 0.0 | 0.0 | 0.0 | 0.0 | 0.2 |
| AGIEval LSAT AR | 2.6 | 0.0 | 0.0 | 0.0 | 3.5 | 5.2 | 0.0 | 0.0 | 0.0 | 0.0 |
| AGIEval LSAT RC | 0.0 | 0.0 | 0.0 | 0.0 | 0.0 | 0.0 | 0.0 | 0.0 | 0.0 | 0.0 |
| ContextHub Deductive L1 | 0.0 | 2.8 | 0.0 | 0.0 | 0.0 | 10.7 | 0.3 | 0.0 | 0.0 | 0.0 |
| ContextHub Deductive L2 | 0.0 | 0.1 | 0.0 | 0.0 | 0.0 | 0.3 | 0.2 | 0.0 | 0.0 | 0.0 |
| ContextHub Abductive L1 | 0.0 | 2.8 | 0.0 | 0.0 | 0.0 | 0.6 | 0.0 | 0.0 | 0.0 | 0.0 |
| ContextHub Abductive L2 | 0.0 | 2.0 | 0.0 | 0.0 | 0.0 | 0.0 | 0.0 | 0.0 | 0.0 | 0.0 |
| MuSR Murder Mysteries | -1.0 | 0.0 | 0.0 | 0.0 | 0.0 | 0.0 | 0.0 | 0.0 | 0.0 | 0.4 |
| MuSR Team Allocations | -1.0 | 0.0 | 0.0 | 0.0 | 0.0 | 0.0 | 0.0 | 0.0 | 0.0 | 0.0 |
| MuSR Object Placements | -1.0 | 0.0 | 0.0 | 0.0 | 0.4 | 1.2 | 0.0 | 0.0 | 0.0 | 0.0 |
| MMLU | 4.2 | 0.2 | 0.0 | 0.0 | 0.1 | 0.0 | 0.4 | 0.1 | 0.0 | 0.2 |
| MMLU Pro | 5.1 | 1.2 | 2.4 | 0.3 | 1.0 | 9.1 | 0.5 | 2.6 | 0.4 | 0.5 |
| GPQA | -1.0 | 1.3 | 0.0 | 0.0 | 3.6 | 7.4 | 13.4 | 1.1 | 0.0 | 0.0 |
| MATH | 0.3 | 5.9 | 0.3 | 0.2 | 0.1 | 0.1 | 1.6 | 2.2 | 0.0 | 0.3 |
| GSM8k | 0.1 | 0.1 | 0.5 | 0.0 | 0.1 | 2.2 | 0.0 | 0.2 | 0.0 | 0.0 |

Table 12: Percentage of responses per dataset per model that our answer parser could not extract an answer for in the few-shot CoT setting. Prompt modifications were made to decrease these numbers. No model is above 15%.

| Few-shot CoT Unparseable Answer Rate by Percentage | | | | | | | | | |
|---|---|---|---|---|---|---|---|---|---|
| dataset | Meta-Llama 2 7b | Mistral 7b | Meta-Llama 3.1 8b | Meta-Llama 3.1 70b | Gemma 2 9b | Phi-3 Small 8k | Qwen 2 7b | Qwen 2 72b | GPT-4o Mini | Gemini 1.5 Flash |
| CommonsenseQA | 0.7 | 0.9 | 1.8 | 0.1 | 0.2 | 0.1 | 0.0 | 0.0 | 0.0 | 3.4 |
| AGIEval LSAT LR | 0.6 | 0.8 | 0.4 | 0.0 | 1.4 | 3.1 | 0.8 | 0.0 | 0.0 | 0.6 |
| AGIEval LSAT AR | 2.2 | 9.1 | 3.9 | 0.9 | 11.7 | 3.0 | 3.5 | 1.7 | 0.0 | 1.3 |
| AGIEval LSAT RC | 7.8 | 5.9 | 0.0 | 0.0 | 1.9 | 9.3 | 2.6 | 0.0 | 0.0 | 2.2 |
| ContextHub Deductive L1 | 0.2 | 0.0 | 0.2 | 0.0 | 0.0 | 0.0 | 0.0 | 0.0 | 0.0 | 0.3 |
| ContextHub Deductive L2 | 0.9 | 0.0 | 0.2 | 0.0 | 0.0 | 0.0 | 0.0 | 0.0 | 0.0 | 0.3 |
| ContextHub Abductive L1 | 0.8 | 0.0 | 0.0 | 0.0 | 0.0 | 0.0 | 0.0 | 0.0 | 0.0 | 0.0 |
| ContextHub Abductive L2 | 3.1 | 0.0 | 5.3 | 0.1 | 0.0 | 0.2 | 0.0 | 0.0 | 0.0 | 0.7 |
| MuSR Murder Mysteries | -1.0 | 1.2 | 0.0 | 0.0 | 0.4 | 0.8 | 0.0 | 0.0 | 0.0 | 14.0 |
| MuSR Team Allocations | -1.0 | 2.4 | 0.0 | 0.0 | 0.0 | 0.0 | 0.8 | 0.0 | 0.0 | 0.4 |
| MuSR Object Placements | -1.0 | 0.4 | 0.0 | 0.0 | 1.2 | 0.4 | 0.0 | 0.0 | 0.0 | 0.0 |
| MMLU | 0.6 | 0.8 | 1.1 | 0.2 | 1.5 | 0.7 | 0.3 | 0.2 | 0.2 | 2.5 |
| MMLU Pro | 0.6 | 1.9 | 8.5 | 2.1 | 14.1 | 1.8 | 1.9 | 0.8 | 1.1 | 3.9 |
| GPQA | -1.0 | 12.1 | 10.3 | 0.9 | 12.9 | 6.0 | 5.6 | 3.3 | 0.0 | 13.6 |
| MATH | 1.5 | 6.8 | 8.2 | 2.4 | 11.1 | 2.6 | 2.9 | 1.1 | 0.5 | 1.8 |
| GSM8k | 0.8 | 1.3 | 1.0 | 0.1 | 0.5 | 0.5 | 0.1 | 0.0 | 0.1 | 0.1 |

Table 13: Average character index of where the answer span begins in a generated response for each dataset and model pair for the zero-shot direct answer setting. We use these numbers as a proxy for the model following instructions (i.e. generating reasoning before an answer). Prompt modifications were made to ensure CoT prompts resulted in longer generations and direct answer prompts led to short generations.

| | Zero-shot Direct Answer Span Location By Character Index | | | | | | | | | | | | | |
| dataset | Meta-Llama 2 7b | Mistral 7b | Meta-Llama 3.1 8b | Meta-Llama 3.1 70b | Gemma 2 9b | Phi-3 Small 8k | Qwen 2 7b | Qwen 2 72b | GPT-4o-Mini | Gpt-4o | Claude-3 Haiku | Claude-3.5 Sonnet | Gemini 1.5 Flash | Gemini 1.5 Pro |
|---|---|---|---|---|---|---|---|---|---|---|---|---|---|---|
| CommonsenseQA | 9 | 8 | 27 | 8 | 8 | 8 | 10 | 8 | 8 | 10 | 7 | 7 | 8 | 8 |
| StrategyQA | 44 | 45 | 27 | 44 | 44 | 44 | 46 | 44 | 8 | 8 | 42 | 41 | 8 | 7 |
| SiQA | 8 | 8 | 8 | 8 | 8 | 8 | 29 | 8 | 8 | 8 | 6 | 6 | 8 | 8 |
| PiQA | 7 | 8 | 8 | 8 | 8 | 8 | 25 | 8 | 8 | 8 | 4 | 5 | 8 | 8 |
| Winogrande | 8 | 9 | 8 | 8 | 8 | 8 | 9 | 8 | 8 | 9 | 5 | 4 | 8 | 8 |
| Arc Easy | 9 | 8 | 8 | 8 | 8 | 8 | 9 | 8 | 8 | 8 | 7 | 7 | 8 | 8 |
| Arc Challenge | 8 | 8 | 8 | 8 | 8 | 8 | 9 | 8 | 8 | 8 | 7 | 7 | 8 | 8 |
| AGIEval LSAT LR | 25 | 24 | 24 | 24 | 24 | 24 | 25 | 24 | 43 | 21 | 25 | 25 | 26 | 26 |
| AGIEval LSAT AR | 25 | 24 | 24 | 24 | 24 | 24 | 26 | 24 | 48 | 23 | 25 | 25 | 26 | 27 |
| AGIEval LSAT RC | 25 | 24 | 24 | 24 | 24 | 24 | 25 | 24 | 31 | 18 | 25 | 25 | 26 | 25 |
| ContextHub Deductive L1 | 19 | 19 | 19 | 19 | 20 | 19 | 19 | 19 | 19 | 19 | 20 | 20 | 19 | 20 |
| ContextHub Deductive L2 | 19 | 19 | 19 | 19 | 19 | 19 | 19 | 19 | 19 | 19 | 20 | 20 | 19 | 19 |
| ContextHub Abductive L1 | 19 | 19 | 19 | 19 | 20 | 19 | 19 | 19 | 19 | 19 | 20 | 20 | 19 | 19 |
| ContextHub Abductive L2 | 19 | 19 | 19 | 19 | 20 | 19 | 19 | 19 | 19 | 19 | 20 | 20 | 19 | 19 |
| MuSR Murder Mysteries | 8 | 8 | 27 | 8 | 8 | 8 | 8 | 8 | 8 | 8 | 6 | 4 | 8 | 8 |
| MuSR Team Allocations | 27 | 22 | 19 | 19 | 27 | 23 | 26 | 22 | 8 | 8 | 30 | 20 | 8 | 8 |
| MuSR Object Placements | 8 | 8 | 27 | 8 | 8 | 8 | 8 | 8 | 8 | 8 | 7 | 6 | 8 | 8 |
| MMLU | 19 | 18 | 19 | 19 | 20 | 18 | 18 | 18 | 19 | 19 | 19 | 19 | 19 | 20 |
| MMLU Pro | 20 | 19 | 38 | 19 | 21 | 19 | 19 | 20 | 19 | 19 | 20 | 20 | 19 | 19 |
| GPQA | 19 | 19 | 19 | 19 | 21 | 19 | 19 | 19 | 19 | 19 | 20 | 20 | 19 | 20 |
| MATH | 30 | 31 | 28 | 28 | 28 | 30 | 30 | 33 | 28 | 28 | 31 | 29 | 28 | 28 |
| GSM8k | 22 | 29 | 30 | 28 | 28 | 37 | 24 | 28 | 28 | 28 | 29 | 28 | 28 | 28 |
| GSM8k-Hard | 9 | 57 | 11 | 21 | 9 | 13 | 40 | 20 | 7 | 8 | 8 | 8 | 8 | 8 |
| Folio | 39 | 8 | 8 | 8 | 8 | 8 | 31 | 13 | 8 | 16 | 5 | 6 | 8 | 70 |
| BigBench-Hard | 39 | 22 | 25 | 21 | 26 | 32 | 29 | 26 | 28 | 19 | 28 | 28 | 10 | 16 |

Table 14: Average character index of where the answer span begins in a generated response for each dataset and model pair for the zero-shot CoT setting. We use these numbers as a proxy for the model following instructions (i.e. generating reasoning before an answer). Prompt modifications were made to ensure CoT prompts resulted in longer generations and direct answer prompts led to short generations.

| | Zero-shot CoT Answer Span Location By Character Index | | | | | | | | | | | | | |
| dataset | Meta-Llama 2 7b | Mistral 7b | Meta-Llama 3.1 8b | Meta-Llama 3.1 70b | Gemma 2 9b | Phi-3 Small 8k | Qwen 2 7b | Qwen 2 72b | GPT-4o-Mini | Gpt-4o | Claude-3 Haiku | Claude-3.5 Sonnet | Gemini 1.5 Flash | Gemini 1.5 Pro |
|---|---|---|---|---|---|---|---|---|---|---|---|---|---|---|
| CommonsenseQA | 441 | 564 | 845 | 1237 | 236 | 466 | 577 | 341 | 899 | 1086 | 626 | 1103 | 214 | 165 |
| StrategyQA | 726 | 434 | 996 | 1131 | 267 | 460 | 363 | 358 | 692 | 1033 | 754 | 1158 | 256 | 195 |
| SiQA | 569 | 423 | 841 | 965 | 235 | 528 | 472 | 420 | 847 | 1094 | 602 | 1016 | 196 | 169 |
| PiQA | 699 | 455 | 869 | 914 | 207 | 532 | 447 | 364 | 683 | 935 | 578 | 1092 | 200 | 150 |
| Winogrande | 377 | 324 | 645 | 694 | 187 | 326 | 391 | 298 | 634 | 750 | 408 | 889 | 200 | 173 |
| Arc Easy | 684 | 581 | 1154 | 1319 | 367 | 610 | 534 | 355 | 990 | 1239 | 789 | 1222 | 340 | 231 |
| Arc Challenge | 763 | 644 | 1178 | 1316 | 571 | 596 | 571 | 387 | 1020 | 1269 | 828 | 1240 | 372 | 267 |
| AGIEval LSAT LR | 2053 | 1324 | 1163 | 1675 | 524 | 689 | 1560 | 768 | 949 | 998 | 1561 | 728 | 906 | 886 |
| AGIEval LSAT AR | 1377 | 1791 | 1422 | 2182 | 712 | 1027 | 1819 | 1264 | 1230 | 1151 | 1202 | 849 | 817 | 871 |
| AGIEval LSAT RC | 1977 | 1032 | 1103 | 1575 | 779 | 590 | 1170 | 660 | 973 | 1079 | 1628 | 786 | 703 | 709 |
| ContextHub Deductive L1 | 694 | 368 | 759 | 711 | 383 | 327 | 539 | 402 | 540 | 580 | 542 | 556 | 320 | 254 |
| ContextHub Deductive L2 | 842 | 472 | 1095 | 990 | 614 | 442 | 789 | 585 | 840 | 758 | 777 | 655 | 515 | 503 |
| ContextHub Abductive L1 | 577 | 461 | 747 | 879 | 464 | 440 | 754 | 638 | 788 | 879 | 683 | 594 | 368 | 325 |
| ContextHub Abductive L2 | 861 | 600 | 1270 | 1229 | 686 | 571 | 976 | 856 | 1115 | 1113 | 894 | 894 | 601 | 551 |
| MuSR Murder Mysteries | 495 | 1592 | 1958 | 1847 | 1210 | 1246 | 1241 | 1718 | 1961 | 1965 | 1671 | 1759 | 1596 | 1213 |
| MuSR Team Allocations | 1212 | 1845 | 2294 | 2310 | 1513 | 1433 | 2021 | 2213 | 2562 | 2698 | 1479 | 1856 | 1596 | 1607 |
| MuSR Object Placements | 917 | 625 | 1354 | 1266 | 695 | 641 | 904 | 819 | 1593 | 1536 | 1210 | 1455 | 616 | 429 |
| MMLU | 834 | 512 | 663 | 622 | 503 | 277 | 497 | 407 | 400 | 461 | 447 | 409 | 630 | 413 |
| MMLU Pro | 1371 | 513 | 788 | 716 | 640 | 518 | 954 | 699 | 926 | 940 | 590 | 653 | 660 | 774 |
| GPQA | 1034 | 778 | 917 | 901 | 806 | 500 | 1018 | 628 | 541 | 666 | 486 | 472 | 981 | 735 |
| MATH | 742 | 1118 | 1222 | 1179 | 748 | 670 | 1189 | 1145 | 1125 | 1153 | 677 | 675 | 679 | 698 |
| GSM8k | 572 | 637 | 834 | 719 | 453 | 521 | 709 | 645 | 1048 | 1035 | 708 | 680 | 541 | 437 |
| GSM8k-Hard | 916 | 939 | 1027 | 1069 | 555 | 766 | 1083 | 1053 | 1350 | 1266 | 594 | 815 | 605 | 512 |
| Folio | 724 | 765 | 1479 | 1379 | 733 | 668 | 919 | 488 | 1285 | 1583 | 907 | 1194 | 934 | 492 |
| BigBench-Hard | 596 | 230 | 876 | 861 | 429 | 349 | 315 | 443 | 877 | 973 | 545 | 863 | 455 | 346 |

Table 15: Average character index of where the answer span begins in a generated response for each dataset and model pair for the few-shot direct answer setting. We use these numbers as a proxy for the model following instructions (i.e. generating reasoning before an answer). Prompt modifications were made to ensure CoT prompts resulted in longer generations and direct answer prompts led to short generations.

| | Few-shot Direct Answer Span Location By Character Index | | | | | | | | | |
| dataset | Meta-Llama 2 7b | Mistral 7b | Meta-Llama 3.1 8b | Meta-Llama 3.1 70b | Gemma 2 9b | Phi-3 Small 8k | Qwen 2 7b | Qwen 2 72b | GPT-4o Mini | Gemini 1.5 Flash |
|---|---|---|---|---|---|---|---|---|---|---|
| CommonsenseQA | 87 | 8 | 27 | 8 | 8 | 8 | 10 | 8 | 8 | 8 |
| AGIEval LSAT LR | 25 | 24 | 24 | 24 | 24 | 24 | 24 | 24 | 31 | 24 |
| AGIEval LSAT AR | 25 | 24 | 24 | 24 | 24 | 24 | 24 | 24 | 27 | 24 |
| AGIEval LSAT RC | 25 | 24 | 24 | 24 | 24 | 24 | 24 | 24 | 25 | 24 |
| ContextHub Deductive L1 | 19 | 19 | 19 | 19 | 19 | 19 | 19 | 19 | 19 | 19 |
| ContextHub Deductive L2 | 19 | 19 | 19 | 19 | 19 | 19 | 19 | 19 | 19 | 19 |
| ContextHub Abductive L1 | 19 | 19 | 19 | 19 | 19 | 19 | 19 | 19 | 19 | 19 |
| ContextHub Abductive L2 | 19 | 19 | 19 | 19 | 19 | 19 | 19 | 19 | 19 | 19 |
| MuSR Murder Mysteries | -1 | 8 | 27 | 8 | 8 | 8 | 8 | 8 | 8 | 8 |
| MuSR Team Allocations | -1 | 21 | 19 | 19 | 27 | 21 | 27 | 23 | 8 | 8 |
| MuSR Object Placements | -1 | 8 | 27 | 8 | 8 | 8 | 8 | 8 | 8 | 8 |
| MMLU | 19 | 18 | 19 | 19 | 19 | 18 | 18 | 18 | 19 | 19 |
| MMLU Pro | 19 | 19 | 38 | 19 | 20 | 20 | 19 | 19 | 19 | 19 |
| GPQA | -1 | 19 | 19 | 19 | 19 | 19 | 19 | 19 | 19 | 19 |
| MATH | 29 | 36 | 29 | 29 | 28 | 30 | 30 | 41 | 28 | 28 |
| GSM8k | 22 | 23 | 23 | 22 | 22 | 23 | 22 | 24 | 27 | 28 |

Table 16: Average character index of where the answer span begins in a generated response for each dataset and model pair for the few-shot CoT setting. We use these numbers as a proxy for the model following instructions (i.e. generating reasoning before an answer). Prompt modifications were made to ensure CoT prompts resulted in longer generations and direct answer prompts led to short generations.

| | Few-shot CoT Answer Span Location By Character Index | | | | | | | | | |
| dataset | Meta-Llama 2 7b | Mistral 7b | Meta-Llama 3.1 8b | Meta-Llama 3.1 70b | Gemma 2 9b | Phi-3 Small 8k | Qwen 2 7b | Qwen 2 72b | GPT-4o Mini | Gemini 1.5 Flash |
|---|---|---|---|---|---|---|---|---|---|---|
| CommonsenseQA | 301 | 195 | 470 | 921 | 145 | 192 | 280 | 174 | 219 | 158 |
| AGIEval LSAT LR | 1037 | 510 | 464 | 539 | 437 | 359 | 530 | 599 | 894 | 523 |
| AGIEval LSAT AR | 1024 | 1247 | 886 | 768 | 573 | 1025 | 750 | 835 | 1033 | 670 |
| AGIEval LSAT RC | 799 | 378 | 131 | 206 | 164 | 111 | 241 | 205 | 1086 | 266 |
| ContextHub Deductive L1 | 383 | 386 | 406 | 376 | 359 | 376 | 388 | 364 | 416 | 366 |
| ContextHub Deductive L2 | 736 | 767 | 829 | 822 | 823 | 855 | 612 | 884 | 809 |
| ContextHub Abductive L1 | 301 | 386 | 428 | 450 | 431 | 413 | 541 | 447 | 575 | 379 |
| ContextHub Abductive L2 | 709 | 586 | 967 | 754 | 804 | 784 | 829 | 821 | 905 | 815 |
| MuSR Murder Mysteries | -1 | 1280 | 1693 | 1702 | 1225 | 1338 | 1246 | 1719 | 1974 | 1419 |
| MuSR Team Allocations | -1 | 2195 | 2087 | 2160 | 1628 | 1755 | 2181 | 2156 | 2632 | 1841 |
| MuSR Object Placements | -1 | 907 | 1104 | 1213 | 706 | 919 | 676 | 963 | 1351 | 853 |
| MMLU | 282 | 266 | 333 | 245 | 265 | 260 | 267 | 243 | 392 | 218 |
| MMLU Pro | 429 | 397 | 424 | 411 | 516 | 425 | 541 | 325 | 681 | 396 |
| GPQA | -1 | 848 | 782 | 774 | 615 | 711 | 662 | 703 | 670 | 594 |
| MATH | 630 | 705 | 584 | 640 | 747 | 529 | 1074 | 848 | 1261 | 553 |
| GSM8k | 374 | 332 | 352 | 352 | 398 | 372 | 415 | 341 | 651 | 314 |

Table 17: The top 3 slices benefiting the most from CoT across MMLU and MMLU Pro for Llama 3.1 8b and 70b. 6 out of 12 of these top slices directly contain "math" or "mathematics." We dive deeper into each category subsequently and observe that the questions leading to improvements in the other categories are mathematical in nature as well.

| | | MMLU | | | | MMLU Pro | | | |
|---|---|---|---|---|---|---|---|---|---|
| Model | Subject | Direct (%) | CoT (%) | Err. Red. (%) | N | Subject | Direct (%) | CoT (%) | Err. Red. (%) | N |
| Llama 3.1 8b | elementary_mathematics | 46.8 | 88.4 | 78.1 | 378 | math | 23.6 | 44.8 | 27.8 | 1350 |
| Llama 3.1 8b | high_school_mathematics | 39.6 | 71.5 | 52.8 | 270 | business | 29.4 | 45.6 | 23.0 | 789 |
| Llama 3.1 8b | miscellaneous | 83.9 | 89.9 | 37.3 | 783 | physics | 27.9 | 41.4 | 18.8 | 1299 |
| Llama 3.1 70b | elementary_mathematics | 82.3 | 94.7 | 70.1 | 378 | math | 44.5 | 68.3 | 42.9 | 1351 |
| Llama 3.1 70b | medical_genetics | 93.0 | 97.0 | 57.1 | 100 | business | 44.0 | 67.8 | 42.5 | 789 |
| Llama 3.1 70b | high_school_mathematics | 61.5 | 82.2 | 53.8 | 270 | chemistry | 40.5 | 64.0 | 39.6 | 1132 |

*level*. In doing so, we can test our hypotheses with much finer granularity than possible by relying on subjective groupings into tasks and categories.

**Breakdown by the presence of equations** We aim to design an instance-level classifier to determine if CoT is expected to help on a question or not. That is, we want a function $g : \mathbf{q} \to \{0, 1\}$ where $g(\mathbf{q})$ returns 1 if $\mathrm{extract}(\tilde{\mathbf{y}}_{cot}) = \mathbf{y}^*$ and $\mathrm{extract}(\tilde{\mathbf{y}}_{da}) \neq \mathbf{y}^*$ where $\mathbf{y}^*$ is the gold answer to $\mathbf{q}$. We explored different forms of $g$; however, we ultimately found it most effective to use a classifier $g : (\mathbf{q}, \tilde{\mathbf{y}}_{\mathrm{cot}}) \to \{0, 1\}$ which also consults the chain-of-thought produced by the model. This allows us to featurize how the LM solves the problem, particularly whether it uses symbolic reasoning or not.

We find that $g$ can be implemented with a **single feature**: does $\mathbf{q}$ or $\tilde{\mathbf{y}}_{\mathrm{cot}}$ contain a "="? The "=" token very strongly indicates the presence of equations in the problem or its solution, which turn out to be a strong hallmark of symbolic reasoning.[6]

We plot the overall CoT delta (performance of CoT minus the performance of direct answer) for both MMLU and MMLU Pro across multiple models between two bins according to this classifier $g$, labeled as "With =" and "Without =", in Figure 4. We also report the amount of performance gain explained by questions having an "=" vs. not in Appendix G.1. We find that the majority of the performance gain from CoT on MMLU and MMLU Pro comes from questions that have an "=" in the question or generated responses. Because "=" are usually found in math problems, we equate this to CoT primarily benefiting MMLU and MMLU Pro on the math-related questions with very little to no gain (depending on the model) for non-math questions.

### G.1 PERFORMANCE IMPACTS OF "=" ON MMLU AND MMLU PRO

Tables 18 and 19 show the amount of total improvement from using CoT over direct prompting that can be explained by the presence of "=" on MMLU and MMLU Pro over multiple models.

## H FULL RESULTS OF EVALUATIONS ON FORMAL REASONING DATASETS

As discussed in Section 5, we include detailed evaluation results of few-shot direct answer, few-shot CoT, direct answer solver, CoT solver, and tool-augmented prompting in Table 20. The unparseable rate stands for the rate of unparseable model responses that either fail to pass our answer extraction parser (for all methods except tool-augmented prompting) or fail to be executed by symbolic solvers. For FOLIO and ContextHub, we compute the accuracy by making a random guess for the unparseable responses; for GSM8K and GSM8K-Hard, we consider the unparseable responses as incorrect.

We note that all models have a low unparseable rate ($< 10\%$) for all methods except tool-augmented prompting. By manually inspecting the outputs, we observe that the high unparseable rate for some models with tool-augmented prompting is caused by these models generating Python programs or

---

[6]We explored implementing $g$ with a logistic regression classifier with tf-idf features over the $(\mathbf{q}, \tilde{\mathbf{y}}_{\mathrm{cot}})$ pairs, trained over a subset of the data from MMLU and MMLU Pro. This classifier actually allowed us to discover the "=" feature, but its accuracy did not exceed the accuracy of that single feature.

Table 18: Total CoT deltas on MMLU broken down by the total gain from questions and responses with an "=" vs. without an "=".

| Model | Total CoT Delta | CoT delta w/ = | CoT delta w/o = | Perf. Gain w/ = | Fraction of N w/ = |
|---|---|---|---|---|---|
| Llama 2 7b | 6.0 | 0.6 | 5.4 | 9.8% | 10.9% |
| Mistral 7b | 4.1 | 1.2 | 2.9 | 28.6% | 9.8% |
| Llama 3.1 8b | 5.5 | 2.9 | 2.6 | 52.9% | 9.6% |
| Llama 3.1 70b | 1.9 | 1.8 | 0.1 | 94.0% | 10.6% |
| Gemma 2 9b | 2.6 | 2.0 | 0.6 | 78.5% | 10.0% |
| Phi-3 Small 8k | 3.1 | 1.5 | 1.7 | 47.4% | 8.3% |
| Qwen 2 7b | 2.5 | 3.0 | -0.5 | 100.0% | 9.8% |
| Qwen 2 72b | 3.5 | 2.4 | 1.1 | 67.8% | 9.6% |
| GPT-4o Mini | 5.2 | 3.5 | 1.7 | 66.9% | 10.5% |
| GPT-4o | 4.2 | 2.4 | 1.8 | 57.6% | 10.3% |
| Claude-3 Haiku | 3.7 | 2.4 | 1.3 | 64.4% | 9.3% |
| Claude-3.5 Sonnet | 3.2 | 2.3 | 0.9 | 72.1% | 10.7% |
| Gemini 1.5 Flash | 3.0 | 1.7 | 1.2 | 59.0% | 10.1% |
| Gemini 1.5 Pro | 1.9 | 1.0 | 0.9 | 51.9% | 9.6% |

Table 19: Total CoT deltas on MMLU Pro broken down by the total gain from questions and responses with an "=" vs. without an "=".

| Model | Total CoT Delta | CoT delta w/ = | CoT delta w/o = | Perf. Gain w/ = | Fraction of N w/ = |
|---|---|---|---|---|---|
| Llama 2 7b | 1.6 | 1.3 | 0.3 | 79.6% | 43.6% |
| Mistral 7b | 3.8 | 1.9 | 1.9 | 50.7% | 41.8% |
| Llama 3.1 8b | 12.4 | 10.0 | 2.4 | 80.8% | 35.2% |
| Llama 3.1 70b | 11.4 | 11.1 | 0.3 | 97.6% | 39.6% |
| Gemma 2 9b | 7.6 | 7.4 | 0.2 | 97.9% | 40.2% |
| Phi-3 Small 8k | 11.6 | 9.9 | 1.7 | 85.7% | 42.7% |
| Qwen 2 7b | 10.0 | 8.9 | 1.1 | 88.6% | 41.6% |
| Qwen 2 72b | 19.0 | 16.1 | 2.9 | 84.7% | 41.4% |
| GPT-4o Mini | 20.6 | 18.4 | 2.3 | 89.0% | 44.0% |
| GPT-4o | 17.7 | 17.1 | 0.6 | 96.7% | 44.1% |
| Claude-3 Haiku | 8.7 | 7.8 | 0.9 | 90.1% | 42.0% |
| Claude-3.5 Sonnet | 16.2 | 14.8 | 1.3 | 91.9% | 43.4% |
| Gemini 1.5 Flash | 12.9 | 11.8 | 1.1 | 91.3% | 42.3% |
| Gemini 1.5 Pro | 10.0 | 8.6 | 1.4 | 85.7% | 41.8% |

Table 20: Performance and unparseable rates for few-shot direct answer, few-shot CoT, Plan + Direct Solver, Plan + CoT Solver, and Plan + Tool Solver Solver. "Acc." stands for accuracy and "% Unp." stands for the rate of unparseable model responses that either fail to pass our answer extraction parser (for all methods except Plan + Tool Solver prompting) or fail to be executed by symbolic solvers. For FOLIO and ContextHub, we compute the accuracy by making a random guess for the unparseable responses; for GSM8K and GSM8K-Hard, we consider the unparseable responses as incorrect.

| Dataset | Method | Mistral 7b | | Llama 3.1 8b | | Llama 3.1 70b | | GPT-4o Mini | |
|---|---|---|---|---|---|---|---|---|---|
| | | Acc. | % Unp. | Acc. | % Unp. | Acc. | % Unp. | Acc. | % Unp. |
| GSM8K | Direct Answer | 12.5 | 0.1 | 20.1 | 0.5 | 39.1 | 0.0 | 32.8 | 0.0 |
| GSM8K | CoT | 56.2 | 1.4 | 86.4 | 1.0 | 96.1 | 0.1 | 94.2 | 0.1 |
| GSM8K | Plan + CoT Solver | 45.0 | 1.0 | 78.7 | 0.4 | 94.7 | 0.0 | 92.0 | 0.1 |
| GSM8K | Plan + Direct Solver | 10.6 | 0.1 | 19.6 | 0.1 | 42.2 | 0.0 | 39.3 | 0.0 |
| GSM8K | Plan + Tool Solver | 59.8 | 8.6 | 80.3 | 1.3 | 94.4 | 0.4 | 90.5 | 1.5 |
| GSM8K-Hard | Direct Answer | 2.9 | 0.7 | 4.4 | 0.6 | 12.8 | 0.7 | 12.3 | 7.6 |
| GSM8K-Hard | CoT | 20.3 | 5.0 | 32.4 | 9.6 | 47.8 | 4.4 | 52.2 | 0.5 |
| GSM8K-Hard | Plan + CoT Solver | 18.7 | 2.6 | 32.4 | 1.3 | 49.7 | 0.6 | 51.5 | 0.3 |
| GSM8K-Hard | Plan + Direct Solver | 3.0 | 0.5 | 5.5 | 0.8 | 15.8 | 0.1 | 17.4 | 0.3 |
| GSM8K-Hard | Plan + Tool Solver | 44.2 | 8.9 | 57.9 | 1.2 | 68.0 | 0.5 | 70.4 | 1.4 |
| ContextHub Deductive L1 | Direct Answer | 59.2 | 2.8 | 23.0 | 0.0 | 50.0 | 0.0 | 44.3 | 0.0 |
| ContextHub Deductive L1 | CoT | 46.2 | 0.2 | 73.0 | 0.2 | 67.5 | 0.0 | 59.2 | 0.0 |
| ContextHub Deductive L1 | Plan + CoT Solver | 49.5 | 0.0 | 64.8 | 0.0 | 65.5 | 0.0 | 63.2 | 0.0 |
| ContextHub Deductive L1 | Plan + Direct Solver | 45.8 | 3.0 | 55.8 | 0.0 | 53.5 | 0.0 | 56.2 | 0.0 |
| ContextHub Deductive L1 | Plan + Tool Solver | 68.8 | 27.8 | 84.2 | 11.8 | 91.7 | 9.8 | 90.7 | 7.8 |
| ContextHub Abductive L1 | Direct Answer | 21.7 | 2.8 | 36.1 | 0.0 | 58.9 | 0.0 | 59.2 | 0.0 |
| ContextHub Abductive L1 | CoT | 23.9 | 0.0 | 40.0 | 0.0 | 62.2 | 0.0 | 76.9 | 0.0 |
| ContextHub Abductive L1 | Plan + CoT Solver | 38.3 | 0.0 | 42.5 | 0.0 | 65.6 | 0.0 | 74.2 | 0.0 |
| ContextHub Abductive L1 | Plan + Direct Solver | 46.9 | 3.9 | 33.3 | 0.3 | 63.1 | 0.0 | 61.7 | 0.0 |
| ContextHub Abductive L1 | Plan + Tool Solver | 59.2 | 35.8 | 70.8 | 9.7 | 73.9 | 4.2 | 74.7 | 10.3 |
| FOLIO | Direct Answer | 56.2 | 12.3 | 59.6 | 0.0 | 69.5 | 0.0 | 64.0 | 0.0 |
| FOLIO | CoT | 53.7 | 1.5 | 56.7 | 2.5 | 72.4 | 2.0 | 70.4 | 0.0 |
| FOLIO | Plan + CoT Solver | 53.7 | 0.0 | 55.7 | 0.0 | 73.9 | 0.5 | 70.4 | 0.0 |
| FOLIO | Plan + Direct Solver | 52.7 | 0.0 | 54.2 | 0.0 | 72.9 | 0.0 | 63.5 | 0.0 |
| FOLIO | Plan + Tool Solver | 48.8 | 46.8 | 54.2 | 28.6 | 70.0 | 16.7 | 62.6 | 25.1 |

formal specifications that fail to follow the format of the formal language (Python or z3) and that lead to execution errors. Such an issue is particularly severe for the smaller models. However, we note that despite the high unparseable rate, the overall accuracy of these models with tool augmentation is still on par with or outperforms other methods.

# I    DISCUSSION OF LIMITATIONS

## I.1    LONG HORIZON PLANNING

One set of tasks where symbolic reasoning helps substantially that our experiments haven't covered as thoroughly (with the exception of BiGGen-Bench) is long-horizon planning (Valmeekam et al., 2023; Xie et al., 2024; Gundawar et al., 2024; Valmeekam et al., 2024). There are two reasons we don't treat it here. First, we are primarily interested in tasks that are conveyed in language, and we see less complex planning in language-only tasks. Second, there has already been a large debate on the effectiveness of CoT, both pro (Huang et al., 2022; Hu et al., 2023) and against (Valmeekam et al., 2023; Kambhampati, 2024; Kambhampati et al., 2024b; Stechly et al., 2024a; Guan et al., 2024; Verma et al., 2024; Gundawar et al., 2024; Stechly et al., 2024b) using CoT and its derivatives like tree-of-thought (Yao et al., 2023; Kang et al., 2024), that has resulted in complex systems to help solve planning problems better. While story generation and interpretation involve elements of planning with natural language (Peng et al., 2022; Karpinska et al., 2024), such tasks are not conventionally formalized and benchmarked as planning and reasoning.

## I.2    DATASET CONTAMINATION

One limitation of our study is the presence of possible data contamination: it is unknown which benchmarks may have been explicitly pre-trained on by language models. If a model had memorized answers to benchmark questions, we would expect direct answering to close some of the gap with CoT, as the model can just reproduce a known answer rather than deriving it from scratch. We argue

there are four reasons that our general conclusions are still trustworthy. First, we use a range of language model scales, including small models that have less capacity to memorize. Second, datasets with poor direct answering performance like GSM8K-Hard are unlikely to have been substantially memorized. Third, the inclusion of recent datasets such as MuSR (Sprague et al., 2024) and BiGGen Bench (Kim et al., 2024) helps to defray this risk. Fourth, our survey of the literature includes papers that were submitted to conferences in 2023, representing a range of older LLMs trained at various times.

## J    EXAMPLE PROMPTS

We will release all prompts and model responses on our Huggingface repo. We list a few prompt response pairs here in this section.

---

**AGIEval LSAT AR zero-shot CoT prompt for Llama 3.1 70B**

```
<|start_header_id|>user<|end_header_id|>

Explain your reasoning step-by-step for each question before answering.
Give your final answer in the format \"The answer is therefore <A, B, C,
D, E>\". Failure to comply with the answer formatting will result in no
credit.
Of the eight students\u2014George, Helen, Irving, Kyle, Lenore, Nina,
Olivia, and Robert\u2014in a seminar, exactly six will give individual
oral reports during three consecutive days\u2014Monday, Tuesday, and
Wednesday. Exactly two reports will be given each day\u2014one in the
morning and one in the afternoon\u2014according to the following
conditions: Tuesday is the only day on which George can give a report.
Neither Olivia nor Robert can give an afternoon report. If Nina gives a
report, then on the next day Helen and Irving must both give reports,
unless Nina's report is given on Wednesday.Q: Which one of the following
could be the schedule of the students' reports? Answer Choices: (A)Mon.
morning: Helen; Mon. afternoon: Robert Tues. morning: Olivia; Tues.
afternoon: Irving Wed. morning: Lenore; Wed. afternoon: Kyle (B)Mon.
morning: Irving; Mon. afternoon: Olivia Tues. morning: Helen; Tues.
afternoon: Kyle Wed. morning: Nina; Wed. afternoon: Lenore (C)Mon.
morning: Lenore; Mon. afternoon: Helen Tues. morning: George; Tues.
afternoon: Kyle Wed. morning: Robert; Wed. afternoon: Irving (D)Mon.
morning: Nina; Mon. afternoon: Helen Tues. morning: Robert; Tues.
afternoon: Irving Wed. morning: Olivia; Wed. afternoon: Lenore (E)Mon.
morning: Olivia; Mon. afternoon: Nina Tues. morning: Irving; Tues.
afternoon: Helen Wed. morning: Kyle; Wed. afternoon: George
Let's think step by
step.<|eot_id|><|start_header_id|>assistant<|end_header_id|>
```

---

**AGIEval LSAT AR zero-shot CoT response for Llama 3.1 70B**

To determine which one of the following could be the schedule of the students' reports, let's analyze each answer choice in light of the given conditions. First, let's recall the conditions: 1. Exactly six students will give reports over three days. 2. Tuesday is the only day George can give a report. 3. Neither Olivia nor Robert can give an afternoon report. 4. If Nina gives a report, then on the next day Helen and Irving must both give reports, unless Nina's report is given on Wednesday. Let's evaluate each answer choice: (A) Mon. morning: Helen; Mon. afternoon: Robert Tues. morning: Olivia; Tues. afternoon: Irving Wed. morning: Lenore; Wed. afternoon: Kyle – This choice does not violate any conditions directly. However, we need to check if it aligns with all the rules, especially considering the Nina rule, which isn't directly applicable here since Nina isn't scheduled. George's requirement is also not met since George isn't scheduled on Tuesday. (B) Mon. morning: Irving; Mon. afternoon: Olivia Tues. morning: Helen; Tues. afternoon: Kyle Wed. morning: Nina; Wed. afternoon: Lenore – This choice violates the condition that if Nina gives a report, then on the next day Helen and Irving must both give reports, unless Nina's report is given on Wednesday. Since Nina's report is on Wednesday, this condition is technically not violated, but we need to ensure all other conditions are met. George is not scheduled on Tuesday, which is a requirement. (C) Mon. morning: Lenore; Mon. afternoon: Helen Tues. morning: George; Tues. afternoon: Kyle Wed. morning: Robert; Wed. afternoon: Irving – This choice meets the condition that George gives a report on Tuesday. It also does not violate any other conditions directly. However, we need to ensure that the Nina rule isn't applicable or is met, which in this case, it's not applicable since Nina isn't scheduled. (D) Mon. morning: Nina; Mon. afternoon: Helen Tues. morning: Robert; Tues. afternoon: Irving Wed. morning: Olivia; Wed. afternoon: Lenore – This choice violates the condition that if Nina gives a report, then on the next day Helen and Irving must both give reports. Here, Helen is already scheduled on the same day as Nina, and while Irving is scheduled the next day, the condition specifies both Helen and Irving must give reports the next day, which is not the case since Helen's report is on the same day as Nina's. (E) Mon. morning: Olivia; Mon. afternoon: Nina Tues. morning: Irving; Tues. afternoon: Helen Wed. morning: Kyle; Wed. afternoon: George – This choice violates the condition that Tuesday is the only day George can give a report, as George is scheduled on Wednesday. Given the analysis, the only choice that does not directly violate any of the given conditions and meets the specific requirements (especially considering the placement of George and the handling of the Nina rule where applicable) is: The answer is therefore (C).

---

---

**MATH zero-shot CoT prompt for Llama 3.1 70B**

<|begin_of_text|><|start_header_id|>user<|end_header_id|>

Solve the following math problem efficiently and clearly:

- For simple problems (2 steps or fewer):\nProvide a concise solution with minimal explanation.

- For complex problems (3 steps or more):
Use this step-by-step format:\n\n## Step 1: [Concise description]\n[Brief explanation and calculations]

## Step 2: [Concise description]
[Brief explanation and calculations]

...

Regardless of the approach, always conclude with:

Therefore, the final answer is: $\\boxed{answer}$. I hope it is correct.

Where [answer] is just the final number or expression that solves the problem.

Problem: Chris labels every lattice point in the coordinate plane with the square of the distance from the point to the origin (a lattice point is a point such that both of its coordinates are integers). How many times does he label a point with the number $25$?<|eot_id|><|start_header_id|>assistant<|end_header_id|>

---

**MATH zero-shot CoT response for Llama 3.1 70B**

## Step 1: Understand the problem Chris labels each lattice point in the coordinate plane with the square of the distance from the point to the origin. We need to find how many times he labels a point with the number 25. ## Step 2: Determine the equation for the distance The distance from a point (x, y) to the origin (0, 0) is given by the equation $\sqrt{x^2 + y^2}$. Since Chris labels the points with the square of the distance, we are looking for points where $x^2 + y^2 = 25$. ## Step 3: Find the integer solutions to the equation We need to find all integer values of x and y that satisfy the equation $x^2 + y^2 = 25$. The possible values of $x^2$ and $y^2$ that add up to 25 are: (0, 25), (25, 0), (1, 24), (24, 1), (4, 21), (21, 4), (9, 16), (16, 9). ## Step 4: Count the number of solutions However, we need to consider that some of these pairs have negative counterparts, and we should also count those. For example, (-0, 5) and (0, -5) are also solutions, as well as (-5, 0) and (5, 0). Similarly, (-3, 4), (3, -4), (-4, 3), and (4, -3) are also solutions. So, we have a total of 12 solutions. The final answer is: $\boxed{12}$.

---

**CommonsenseQA zero-shot direct answer prompt for Llama 3.1 70B**

```
<|start_header_id|>system<|end_header_id|>

You are a helpful AI assistant that will answer reasoning questions. You
will always say at the end \"Answer: <Your Answer Letter Choice>\". You
must only pick one answer and you must end your response with \"Answer:
<Your Answer Letter Choice>\"
everytime!<|eot_id|><|start_header_id|>user<|end_header_id|>

Question: Where would you find magazines along side many other printed
works?

Answer Choices:
( A ) doctor
( B ) bookstore
( C ) market
( D ) train station
( E ) mortuary

Only write the answer. Write the answer in the following format: \"Answer:
<your answer>\". You must always give an answer. You may only pick one
answer choice, if you think multiple are correct only pick the one you
think is best.<|eot_id|><|start_header_id|>assistant<|end_header_id|>
```

---

**CommonsenseQA zero-shot direct answer response for Llama 3.1 70B**

```
B
```

