# OpenReview forum: "To CoT or not to CoT? Chain-of-thought helps mainly on math and symbolic reasoning"
_ICLR.cc/2025/Conference — ICLR 2025 Poster_

### Official Review · Reviewer_FSj2 · 2024-11-03

**Soundness:** 3
**Presentation:** 2
**Contribution:** 2
**Rating:** 6
**Confidence:** 4

**Summary:**

The paper presents a comprehensive analysis of the CoT prompting technique. Through a meta-analysis of existing literature and extensive experiments on various datasets and models, the authors demonstrate that CoT is particularly effective for mathematical and symbolic reasoning tasks but offers minimal benefits for other types of reasoning. The study provides valuable insights into the conditions under which CoT excels and suggests a shift towards more advanced computational paradigms for LLMs.

**Strengths:**

1. The paper offers both quantitative performance metrics and qualitative insights into why CoT is effective in certain tasks, enhancing the depth of understanding.

**Weaknesses:**

1. I believe the core findings of this paper are already well-known among many researchers in this field. Although the authors conduct a comprehensive investigation, the core contributions appear to be limited.
2. The authors define symbolic and non-symbolic reasoning but categorize experimental datasets into multiple types, which may lead to confusion.

**Questions:**

In some tasks, the use of CoT reasoning can decrease performance. What do the authors consider to be the key reasons for this? Will this phenomenon continue to be an issue in future LLMs?

---

> ### Author Response · Authors · 2024-11-20
> **Response to reviewer FSj2**
>
> Thank you for your feedback!
>
> > I believe the core findings of this paper are already well-known among many researchers in this field. Although the authors conduct a comprehensive investigation, the core contributions appear to be limited.
>
> Our contributions may be intuitive to some, but from our meta-analysis, we’ve shown that many researchers in the field do not have this understanding of CoT and our findings, according to their recent papers. 11% of the papers released in ICLR, EACL, and NAACL 2024 mention CoT (Appendix A shows more details about these numbers).  We found, perhaps surprisingly, that most of these papers were using CoT for non-symbolic tasks.  This is the motivation behind our paper and our contribution.  We hope that our work shows the limitations of CoT and encourages researchers to find new methods for solving reasoning tasks that do not involve math and logic.
>
>
> > In some tasks, the use of CoT reasoning can decrease performance. What do the authors consider to be the key reasons for this? Will this phenomenon continue to be an issue in future LLMs?
>
> This is a great question! A recent paper actually addresses this question, [Mind Your Step (by Step): Chain-of-Thought can Reduce Performance on Tasks where Thinking Makes Humans Worse](https://arxiv.org/abs/2410.21333?ref=upstract.com).  However, many of the tasks represented in this paper are a bit out of distribution for an LLM (like facial recognition).
>
> In our meta-analysis, we found some examples where CoT performs worse than direct answering when the task is adversarial or specifically designed to test rule inversions (https://aclanthology.org/2024.naacl-long.310.pdf) or when a highly specialized skill is being tested on a general purpose model (https://aclanthology.org/2024.clinicalnlp-1.15.pdf). Some very old tasks also see that CoT hurts performance (https://aclanthology.org/2024.naacl-long.122.pdf).  One theory we had was that for some tasks, direct answering is more “in-distribution” of the pretraining data, and CoT may push the LLM into a space where it’s less sure, so the model may have to overfit to some of these tasks.  More analysis is needed to better understand why intermediate generations are hurting these kinds of tasks.
>
> However, we are hopeful that with improvements in training and architecture, we can get LLMs to reason and effectively improve their performance through increased computation (though this may not be through intermediate generation such as CoT).
>
>
> > The authors define symbolic and non-symbolic reasoning but categorize experimental datasets into multiple types, which may lead to confusion.
>
> Thank you for this feedback. The reason for this is that the definition of what makes something symbolic vs non-symbolic can be difficult to put into practice.  Instead, we tend to see tasks at a more granular level (categories) and then report over those categories, which was a more natural dimension for categorizing datasets and allows for better post hoc analysis of dataset categories than just having them categorized as “symbolic” and “non-symbolic” would.

---

### Official Review · Reviewer_NZkQ · 2024-11-03

**Soundness:** 4
**Presentation:** 4
**Contribution:** 4
**Rating:** 8
**Confidence:** 3

**Summary:**

The paper presents a comprehensive analysis of prompt-based chain-of-thought across a variety of tasks, specifically as compared to direct answer prompting. The analysis proceeds in three parts.

## Part 1
The paper presents a meta-analysis of performance gains due to prompt-based chain-of-thought reported in the literature. They choose a set of 110 papers from recent NLP and ML venues that explicitly compare direct answer prompting to chain-of-thought. They manually extract results from these papers. They they group tasks into 14 categories, and analyze the average performance gain due to chain-of-thought in each category. They find that the top 3 categories – symbolic reasoning, math, and logical reasoning – benefit substantially, while other categories see only a small benefit. There is some discussion of exceptions to this rule, but the trend from the literature suggests that chain-of-thought largely benefits only mathematical or symbolic reasoning tasks.

## Part 2
The paper presents direct comparisons of direct answering, zero-shot, and few-shot chain-of-thought on 14 LLMs on 20 datasets, broadly categorized as Commonsense, Knowledge, Symbolic, Mathematical, and Soft Reasoning. Here too, they find that performance gains are primarily in tasks that require some form of symbolic or mathematical reasoning, with trends being fairly stable across models. Even gains on Commonsense, Knowledge, and Soft Reasoning datasets can be tied back to mathematical reasoning, like in a mixed dataset such as MMLU (where they find that a majority of the gain is in problems that involve the "=" sign).

## Part 3
The paper then examines why chain-of-thought might be so particularly helpful for mathematical or symbolic reasoning. For this, the paper identifies the chain-of-thought process as consisting of two processes – planning and execution. In the planning phase, the problem is mapped to a plan, and in the execution phase, the plan is executed to get an answer. The paper presents a way to compare the influence of these stages by having models generate a plan in code/logic, and then comparing the execution of the plan with direct LLM answering, execution with chain-of-thought, and execution with a symbolic engine. The paper argues that most of the gains of chain-of-thought appear to be from more reliable execution, rather than superior planning.

Given these, the paper concludes that:
- prompt-based chain-of-thought is not an effective method for reasoning construed broadly
  - The paper does highlight that methods that leverage more compute or search, might be more promising, as might methods that train models to use chain-of-thought
- in tasks where chain-of-thought does provide an advantage in better executing complex plans, there may be more reliable symbolic execution options that could do even better

**Strengths:**

- The paper presents a meta-analysis of work in ML and NLP that is appropriate (provides evidence of where chain-of-thought helps without re-running every single experiment)
  - This is not a commonly used approach in ML work, so I cannot be entirely certain, but the methodology of the meta-analysis seems mostly sound.
- The paper reinforces the findings of the meta-analysis with direct comparisons of multiple models across a number of tasks, both complementing and extending the findings of the meta-analysis
  - This comparison seems sound and thorough, particularly with attention to detail such as ensuring answers appear earlier in the response for direct answering as compared to chain-of-thought, which is a simple and effective way to measure that a model does produce a chain-of-thought
  - Identifying the "=" sign as a crucial determiner of gains in MMLU is also a cleverly designed experiment with informative findings
- The design of the study to separate execution and planning is an interesting and useful approach, and uses prior work like PAL in a clever way to set up a controlled study
  - The discussion of how this relates to early work on chain-of-thought (Nye et al., 2021) provides useful context

**Weaknesses:**

- There is no discussion of model size in the meta-analysis. Work that proposed chain-of-thought (such as Wei et al. (2022) and Kojima et al. (2022)) explicitly discuss how size is a crucial factor in the effectiveness of chain-of-thought. In fact, if we consider Table 26 of Kojima et al. (2022) that compares chain-of-thought and direct answering across a range of model sizes, of the 4 models tested, only 1 shows a substantial gain with chain-of-thought. Given that the results presented aggregate across paper and category, and that models of different sizes are not distinguished in the meta-analysis figures, it is unclear how much results of chain-of-thought on smaller models (even as large as 7B (Kojima et al., 2022; Table 26)) is the reason why the method appears to be ineffective.
  - This concern is mostly addressed in the direct comparisons, which show similar findings across a range of sizes, and hence I don't think this significantly bears on my rating, but if something can be said about the role of size in the meta-analysis it might further bolster the argument.
- Section 5 ends with "When possible, LLMs should be paired with symbolic solvers at inference time when solving symbolic tasks to achieve consistently better performance over direct answer **and** CoT." While it is an important takeaway that in some cases, the benefits of chain-of-thought can be reaped to a greater degree with exact symbolic execution, one aspect of the problem that doesn't seem to be discussed is the difficulty of getting models to produce the inputs to symbolic execution engines. These engines are powerful, but brittle, and minor errors in producing a program to be executed can lead to complete failure. Chain-of-thought on the other hand allows the model to be more flexible and robust. This is directly visible in the results, for example, for FOLIO. Models with chain-of-thought or plan+chain-of-thought execution perform _better_ than the plan+tool solver counterparts, which could be largely due to the relatively high unparseability rates. This suggests that the program writer-solver pair of (LLM, LLM) is better than the pair of (LLM, tool solver) in some cases (i.e. the LLM is a better executor of the plans it writes than a tool). If what matters to us is the final answer, should we not consider a solution that makes it more likely to get the correct final answer (setting compute requirements aside for a second)?
  - I'm not suggesting that we should declare chain-of-thought the winner here, just that I think this adds some nuance to what is a definitive recommendation made in the paper. I don't think this would change my recommendation, but I'd be glad if the authors would engage about this.

**Questions:**

> [323] Not much. Free response capabilities may be hindered by pre-planning or reasoning about the correct response.

I'm unsure what this means. Where is the hinderance to free-response capabilities that is being referred to? I might have missed something, but it was unclear from the writing.

> Informally, we believe many readers of the literature to hold the following view: [...]

This may not be a really substantive question, but is there a reason for framing this as a belief held by readers, rather than a specific reference to some prior paper(s) as is the norm?

---

> ### Author Response · Authors · 2024-11-20
> **Response to reviewer NZkQ**
>
> Thank you for your feedback on our paper. We were excited to hear you call the MMLU analysis as clever and the neurosymbolic analysis as useful!
>
> Weakness:
> >There is no discussion of model size in the meta-analysis. [...] This concern is mostly addressed in the direct comparisons
>
> We will release the data for our meta-analysis along with code for a future version of the paper. We do not discuss model sizes for the meta-analysis due to many models of different generations and capabilities being used. It would be difficult to make conclusive statements about scale due to newer, smaller models being better at CoT than models like text-davinci-001, for example. This is in part why we follow up with the direct comparisons as you’ve pointed out. Based on that analysis, it appears that when a model has the ability to perform CoT (which most modern models with 1B+ parameters do), CoT will give benefits on symbolic questions but not so much on non-symbolic questions.
>
> Weakness:
> > one aspect of the problem that doesn't seem to be discussed is the difficulty of getting models to produce the inputs to symbolic execution engines. [...] Chain-of-thought on the other hand allows the model to be more flexible and robust. This is directly visible in the results, for example, for FOLIO. Models with chain-of-thought or plan+chain-of-thought execution perform better than the plan+tool solver counterparts, which could be largely due to the relatively high unparseability rates. [...] If what matters to us is the final answer, should we not consider a solution that makes it more likely to get the correct final answer (setting compute requirements aside for a second)?
>
> This is a great observation and a good point. We agree that we ultimately care about the final answer and that the results on FOLIO and GSM8k seem to favor using only CoT without any symbolic solver. However, we believe that dataset contamination and difficulty of the problems play a role in the results for FOLIO and GSM8k. First, FOLIO and GSM8k are popular datasets and are potentially leaked into the pre-training data (though we do not have direct evidence of this). Second, ContextHub and GSM8k-HARD, which are datasets that contain questions that are very similar but more challenging to FOLIO and GSM8k, see major wins from using a symbolic solver.  We believe this means that as problems become more out of distribution and/or more challenging, you are more likely to get the right final answer by using a symbolic solver, specifically because it may be easier to specify the input to a symbolic solver than to answer the question via the LLM.  For ContextHub, we had an unparseability rate of 10-20% (depending on the model) but still managed to outperform CoT-only variants for all models except one. It may actually be easier to correct these unparseables than to train an LM to solve these complex reasoning questions.
>
>
> Question:
> > “Not much. Free response capabilities may be hindered by pre-planning or reasoning about the correct response.” I’m unsure what this means [...]
>
>
> Thank you for pointing this out; we are making a revision to make this point clearer, and the text will be highlighted so it is easy to spot where we made the change. The free-response capabilities we were discussing were from BiGGen Bench, which requires a language model to answer an open-ended question with a free response. Keeping free responses separate from CoTs (i.e., generating a CoT *and* a long-form response, then deleting the CoT) is not very standard. The free-response is then graded by an LLM-as-a-judge (in our case, GPT-4o-mini).  Despite being free responses, we believe that many of the questions in BiGGen Bench involve symbolic reasoning, and so should receive a benefit from CoT.  However, we only saw a minor improvement on average: some models do get a larger boost, but many get smaller boosts or decline in performance.
>
> Question:
> > Informally, we believe many readers of the literature to hold the following view: [...]
> This may not be a really substantive question, but is there a reason for framing this as a belief held by readers, rather than a specific reference to some prior paper(s) as is the norm?
>
> One answer to this question comes from the meta-analysis itself. Over 11% of the papers published in ICLR, NAACL, and EACL of 2024 mention CoT.  Many of those papers use CoT as a baseline or method for solving a task.  Through our meta-analysis we found that many of these tasks were not symbolic, which is some indirect evidence that people hold the view that CoT should be better than Direct Answer. We will try to draw this connection more clearly in a future version.

---

> > ### Comment · Reviewer_NZkQ · 2024-11-23
> >
> > > We will release the data for our meta-analysis along with code for a future version of the paper.
> >
> > Yes, I think data for meta-analysis that includes the model ID will do more than address the weakness I raised. My concern was less about analyzing size trends, than the possibility of the meta-analysis being based mostly on models that aren't capable of leveraging CoT. Again, this is directly addressed through the controlled comparisons in the paper so is only a minor concern, and having the data of which models the meta-analysis numbers were based would allow people to analyze the data further if they wish.
> >
> > > We believe this means that as problems become more out of distribution and/or more challenging, you are more likely to get the right final answer by using a symbolic solver, specifically because it may be easier to specify the input to a symbolic solver than to answer the question via the LLM.
> >
> > I agree, I think symbolic solvers are more predictable (for now) than LLM solvers, and it is totally reasonable to conclude that this is where there value lies. The additional points you raised about unparseability in ContextHub are also valuable. If space permits, I feel a more nuanced discussion of this point (as the authors have done in their response) would address my concern and strengthen the paper.
> >
> > >  Through our meta-analysis we found that many of these tasks were not symbolic, which is some indirect evidence that people hold the view that CoT should be better than Direct Answer. We will try to draw this connection more clearly in a future version.
> >
> > Yes, my point was less about the claim than the way it was framed ("Informally, [...]", when there are concrete bits of evidence available), so this can be addressed in the writing. In any case, this was more a question than a suggestion to make a change.
> >
> > Overall, my initial rating reflected that the paper was above the bar for acceptance and that holds despite the points I raised. The changes discussed here would be nice to see in the paper, but they aren't critical and can be made for the camera-ready.

---

### Official Review · Reviewer_nciZ · 2024-11-04

**Soundness:** 3
**Presentation:** 4
**Contribution:** 2
**Rating:** 6
**Confidence:** 4

**Summary:**

In this work, the author study the effect of CoT on a wide variety of problems and try to identify categories of problems for which it work v/s ones for which it doesn't. Using their meta-analysis of results from various recent papers and their own analysis spanning models and datasets, they conclude that CoT is more useful for math and logic based tasks, while providing minimal gains for other task tasks.

**Strengths:**

1. CoT is a popular and often over-used strategy in LLM based applications. This paper dives deep into where it actually is useful and where it isn't - contributing to a better overall understanding of the technique and lay foundations for future improvements.
2.  The paper is well-written, detail experiments well and supports its key insights with exhaustive empirical analysis.
3. The extensive analysis of contemporary literature is quite interesting and one of a kind.

**Weaknesses:**

1. While the paper comes up with some interesting analysis, it doesn't really propose alternatives - it claims that CoT is a general approximation to logical solvers but proposes using them during inference. Arguably, it is hard to come up tool augmented neuro-symbolic techniques that fit most symbolic reasoning tasks well -making CoT especially appealing. Even though, this doesn't bring down the paper's contributions, it would have made the work more interesting for me.
2. The paper distinguishes the 264 datasets on which it bases its main position with but doesn't specify how it comes up with the categorization. A detailed explanation of these exact considerations can help better utilize the paper's recommendations in practice.
3. Language models are not a monolith and different Language Models behave differently depending on the fine-tuning dataset, preference alignment etc. Trends in Fig 6 already highlight the variation across LMs. The paper makes broad claims without often specifying the LM over which the result were found- almost assuming that they will generalize across models. A few examples:
	* Figure 3: What are the 5 representative models?
	* Line 320-321: Analysis of the impact of few-shot CoT v/s direct ?
Besides, the paper lacks key model-wise analysis of its findings.

**Questions:**

1. What are the considerations the authors took to categorize the datasets into the proposed categories?
2. How does the paper's findings hold across LLMs?
3. Minor:
    * What are the 5 representative models used in Figure 3 right?
    * What do different colors in Figure 4 represent?

---

> ### Author Response · Authors · 2024-11-20
> **Response to reviewer nciZ**
>
> Thank you for your feedback, we are excited to hear you describe our analysis as “a one of a kind”!  We also agree that CoT is being used significantly throughout the field and that a better understanding of CoT (and similar methods) could help lay foundations for future improvements.
>
> > What are the considerations the authors took to categorize the datasets into the proposed categories?
>
> A list of definitions for each category can be found in Table 2 of the Appendix.  To determine the categories, we had authors (3) annotate each paper with their own categorization labels.  We then performed an adjudication step in cases of disagreement, which led to refined categories.  We also have all the raw data from our analysis, including the paper, venue, dataset, category (our label), the model they evaluated on, as well as performance metrics.  All of these will be released with the paper.
>
>
> > How does the paper's findings hold across LLMs?
>
> For our finding on CoT only helping on symbolic questions:
> We find that it holds well.  Our meta-analysis, section 3, took papers that were evaluated across many LLMs across sizes, age (Llama 1 / PALM 540 B / GPT-4 etc.) as well as open and closed source models. More detailed data from our meta-analysis will be released subsequently, including the model for each experiment.
>
> Furthermore, our own experiments in Section 4 look at 14 models.  We list them on the x-axis of Figure 3 (left).  We intentionally picked many flagship closed-source models, including GPT-4o, Claude-3.5 Sonnet, and Gemini 1.5 Pro, as well as weaker open-source models like Llama 2 7B, Qwen 2 72B, etc. to help establish the trend we report in our paper across models.
>
> For our finding on CoT being worse than symbolic solvers
> For this experiment, we took a smaller pool of 4 models we believe represent a good sample.  We list them in Figure 6 in section 5.1.  They are Mistral 7B, Llama 3.1 8B, Llama 3.1 70B, and GPT-4o Mini.
>
> > What are the 5 representative models used in Figure 3 right?
>
> The models are mentioned in the title of each graph.  “Averaged” is across all 14 of our models.  The rest are a sample of individual models. They are, in order from left to right, Qwen 2 7b, Llama 3.1 8B, Llama 3.1 70B, GPT-4o, and Claude-3.5 Sonnet.
>
>
>
> > What do different colors in Figure 4 represent?
>
> From the legend, MMLU Pro is colored as blue whereas MMLU is colored purple.  Hatch patterns (slightly diluted coloring) indicate a subset of each dataset where the question or generated response includes an “=” in them (hence the legends title “MMLU Pro Without =” would be the solid blue bar and “MMLU With =” would be the purple bar with hatching).
>
> MMLU and MMLU Pro get benefits from CoT on their questions that involve math (we use the presence of an “=” as a heuristic for the questions involving math). We believe this analysis is valuable as MMLU and MMLU Pro have been used, and currently are used, to champion the notion that CoT and intermediate generations can help on non-symbolic datasets.  We found in our analysis that MMLU and MMLU Pro are more symbolic than most people think and CoT primarily helps on that symbolic part of the dataset.
>
> > Arguably, it is hard to come up tool augmented neuro-symbolic techniques that fit most symbolic reasoning tasks well -making CoT especially appealing.
>
> This is a good point that Reviewer FSj2 also raises. Please refer to our response to Reviewer FSj2 for our justifications of why tool-augmented neuro-symbolic techniques could be more effective and appealing in practice than CoT.

---

> > ### Comment · Reviewer_nciZ · 2024-12-03
> > **Response to Authors**
> >
> > Thanks for your detailed response. I figure I may have overlooked a few details that made me incorrectly point out a part of Weakness 3, and I thank the authors' for politely pointing the right part of the text for the same.
> > The Authors have also satisfactorily addressed my comment on dataset classification. I will highly recommend the authors to add these details of the methodology in the paper( or the appendix) as well.
> >
> > I still feel that the paper needs more analysis on how size/style of the LLMs impact the performance with and without CoT. I also couldn't find a satisfactory answer for Weakness 1.
> >
> > While I appreciate the merit of the work, given the above concerns, I will keep my initial score unchanged for now.

---

### Meta-Review · Area_Chair_4C4k · 2024-12-23

**Metareview:**

This paper presents a comprehensive analysis of chain-of-thought reasoning using large language models. First, the authors present a meta-analysis of existing literature, consisting of 110 papers, and conclude that CoT reasoning mostly benefits tasks in symbolic/logical reasoning and math. Second, the authors performed direct comparisons of CoT versus direct answering on datasets using 14 models with similar findings. Finally, to further analyze the reason for the effectiveness of CoT on reasoning tasks, the authors decoupled CoT trajectories into a planning and an execution phase, and designed controlled experiments to show that most gains from CoT appears to result in more reliable execution, rather than superior planning.

**Strengths:**

* This paper contributes “to a better overall understanding of the technique (CoT)” (nciZ)
* Reviewers found that the meta-analysis of many work in ML and NLP “seems mostly sound” (NZkQ), “quite interesting” (nciZ), and “one of a kind”.
* All reviewers agree that the experiments are thorough (NZkQ) and exhaustive (nciZ), providing valuable insights (nciZ, FSj2) into the scenarios where CoT is mostly effective.
* “The design of the study to separate execution and planning is an interesting and useful approach” (NZkQ)

**Weaknesses:**

There were some minor issues with the methodology of the meta-analysis, such as how model size might impact the conclusion (NZkQ) and the motivation of the categorization of the datasets chosen (nciZ), which are addressed during the rebuttal phase.

Reviewer NZkQ raises an interesting point about the potential limitations of using symbolic solvers in tandem with LLMs, that it might be non-trivial for LLMs to generate the correct input to the solver, which challenges the author's claim that “when possible, LLMs should be paired with symbolic solvers at inference time when solving symbolic tasks to achieve consistently better performance …”. The authors provided a reasonable explanation in the rebuttal that symbolic solvers would be more effective as the problem complexity grows.

The only remaining issue after the rebuttal is that the paper “doesn't really propose alternatives” to CoT (nciZ), and that “core findings of this paper” (that CoT is only effective for reasoning-heavy tasks) “are already well-known” (FSj2). However, the reviewer also acknowledges that “this doesn't bring down the paper's contributions” (nciZ).

This is a well-written paper with extensive literature review, thorough experiments, interesting insights and findings. Given this, the decision is **Accept**.

**Additional Comments On Reviewer Discussion:**

Please refer to the meta review.

---

### Decision · Program_Chairs · 2025-01-22

Accept (Poster)